# *Caenorhabditis elegans* foraging patterns follow a simple rule of thumb

Gabriel Madirolas [1,4], Alid Al-Asmar[1,4], Lydia Gaouar[1], Leslie Marie-Louise[1], Andrea Garza-Enríquez[1], Valentina Rodríguez-Rada[1], Mikail Khona[2], Martina Dal Bello [2], Christoph Ratzke[2,3], Jeff Gore [2] & Alfonso Pérez-Escudero [1,2 ✉]

Rules of thumb are behavioral algorithms that approximate optimal behavior while lowering cognitive and sensory costs. One way to reduce these costs is by simplifying the representation of the environment: While the theoretically optimal behavior may depend on many environmental variables, a rule of thumb may use a smaller set of variables that performs reasonably well. Experimental proof of this simplification requires an exhaustive mapping of all relevant combinations of several environmental parameters, which we performed for *Caenorhabditis elegans* foraging by covering systematically combinations of food density (across 4 orders of magnitude) and food type (across 12 bacterial strains). We found that worms' response is dominated by a single environmental variable: food density measured as number of bacteria per unit surface. They disregard other factors such as biomass content or bacterial strain. We also measured experimentally the impact on fitness of each type of food, determining that the rule is near-optimal and therefore constitutes a rule of thumb that leverages the most informative environmental variable. These results set the stage for further investigations into the underlying genetic and neural mechanisms governing this simplification process, and into its role in the evolution of decision-making strategies.

[1] Centre de Recherches sur la Cognition Animale (UMR5169), Centre de Biologie Intégrative, Université de Toulouse, CNRS, UPS, Toulouse 31062, France.
[2] Physics of Living Systems Group, Department of Physics, Massachusetts Institute of Technology, Cambridge, MA, USA. [3] Interfaculty Institute for Microbiology and Infection Medicine Tübingen (IMIT), Cluster of Excellence EXC 2124 "Controlling Microbes to Fight Infections" (CMFI), University of Tübingen, Calwerstrasse 7/1, 72076 Tübingen, Germany. [4]These authors contributed equally: Gabriel Madirolas, Alid Al-Asmar. ✉email: alfonso.perez.escudero@gmail.com

Sophisticated and highly optimal outcomes of animal behavior often emerge from simple rules, called rules of thumb[1–8]. For example, patch-leaving in parasitoid wasps seems to be adaptive regarding multiple indicators of patch and environment quality[7], but this decision may be driven by a simple mechanism: One internal variable that decreases linearly with time and increases sharply whenever the wasp finds a host. The wasp leaves a patch when this variable reaches a threshold[8]. While being easy to implement, this rule yields near-optimal responses[7,8]. Identifying these rules of thumb is key to link the neural and mechanistic implementation of animal behavior to the selective pressures that shape it[9].

Most rules of thumb rely on a simplified internal representation of the environment. For example, optimal food choice may require considering simultaneously many variables such as the spatial distribution of the food sources, their density, their composition in terms of numerous nutrients, etc. Processing all these variables separately is costly, so a rule of thumb may disregard the less informative variables and combine the rest into one or a few quantities, which constitute the internal representation of the environment and will determine the decision. While numerous studies identify variables that dominate behavior[10], demonstrating a simplified internal representation requires showing that any combination of environmental variables that leads to the same internal representation produces the same response. This is challenging, first because behavioral experiments tend to have a large variability which may hide small effects, and second because a convincing proof must test systematically a large number of equivalent combinations. Reaching at the same time a high number of combinations and a sufficient number of replicates to obtain highly accurate average behavior is beyond the experimental throughput in most behavioral experiments.

To address these challenges, we developed a high-throughput pipeline to study the foraging behavior of the nematode *Caenorhabditis elegans*. We focused on foraging (i.e. search and exploitation of food) because it has a clear impact on fitness, the degree of success is relatively easy to measure (in terms of rate of food consumption), and it is thoroughly studied from a theoretical point of view[11]. Thanks to *C. elegans'* high offspring number and small size, we could perform experiments with more than 20 000 age-synchronized individuals in more than 2 000 experimental arenas. Besides allowing for high experimental throughput, *C. elegans'* small nervous system ( ~ 300 neurons), makes it an ideal candidate to implement simple rules of thumb, while its foraging behavior is complex enough to implement the basic elements of optimal foraging, which can be observed for example in its exploration[12–19], learning[20–23], and feeding[24–35] behaviors.

We systematically characterized *C. elegans'* response to food, covering all relevant combinations of food density (across 4 orders of magnitude, from starvation to rich environment) and food composition (across 12 different bacterial strains, from 11 diverse species). Different bacterial strains differ in their composition in terms of many different molecules, as well in their size, shape and mechanical characteristics, encompassing a high number of variables. Despite this high degree of complexity, *C. elegans'* response to all our bacterial strains followed a simple universal trend.

## Results

### *C. elegans'* reaction to food density follows a sigmoidal trend.
Our experimental setup consisted of round agar plates with 5 food patches of different densities, arranged as a regular pentagon (Fig. 1a). *C. elegans* eats bacteria, and each food patch was a drop of bacterial culture whose density had been carefully adjusted by measuring its optical density (OD). The food patches were placed on the day before the experiment, and we ensured that bacterial density remained unchanged by preparing the agar plates without nutrients and with a low dose of bacteriostatic antibiotics. On the day of the experiment, young adult (48-h old) worms were placed at the center of the plate, equidistant to all food patches, and freely explored the environment for 2 h, a time that was short enough to prevent significant food depletion, but long enough for patch occupancy to be roughly constant at the end of the experiment (Supplementary Fig. 1). We placed ~10 worms per plate, and discarded any plates with more than 20 worms. We kept this low number of individuals per plate to prevent food depletion, and also to prevent collective effects such formation of clusters[36,37], or networks[38], which require higher worm densities. To ensure repeatability and sufficient throughput, both the food patches and the worms were placed on the plate by a pipetting robot (see Methods for further details).

A key challenge was to study a wide enough range of bacterial densities, because placing patches of very different densities on the same plate leads to noisy data: Worms accumulate on the high-density patches, leaving very few individuals to assess the low-density range. We solved this challenge by performing several experiments covering smaller and overlapping density ranges. For visualization purposes, we normalized the number of worms in each experiment with respect to a virtual reference to obtain a relative number of worms comparable across conditions (Supplementary Figs. 2, 3 and Methods).

We found that the relative number of worms at a food patch increases with its bacterial density following a sigmoidal trend, best visualized on a double-logarithmic plot (Fig. 1b). While our results are qualitatively comparable to previous studies[19,30,32,34], our high throughput enabled a quantitative description and revealed that *C. elegans'* preference is well described by a mathematical formula with three parameters (Fig. 1c; Eq. 1 in Methods): The height of the sigmoid (called $H$ and defined as the ratio between the number of worms at the high and low density extremes), the speed at which the number of worms increases with bacterial density (called $k$ and defined as the slope at the sigmoid's midpoint), and the bacterial density at which the sigmoid starts to grow (called $D_{attract}$ and defined as the density at which the number of worms reaches 5-fold the number of worms found at patches with zero bacterial density). We can understand this last parameter as the minimum bacterial density needed to attract the worms significantly, so we called it Attraction Density. After fitting these three parameters, our sigmoid describes the experimental data (Fig. 1b, black line).

### *C. elegans'* response follows the same trend for all bacterial strains.
To investigate whether our results are applicable to different types of food, we performed our experiments with 12 different strains, distributed across 11 species and 7 families. Five of these strains are common in *C. elegans* studies, while the remaining 7 strains are bacteria that we isolated from the gut of *C. elegans* worms grown in the lab on different types of natural compost (see Methods).

We found that the responses of *C. elegans* to all bacterial strains can be described by our sigmoidal equation, after fitting its three parameters independently for each strain (Fig. 1d and Supplementary Figures 4, 5). To quantify the overall goodness of this fit, we compared the proportion of worms at each food patch for each experimental condition with the predictions of our sigmoidal model (Supplementary Fig. 2b). We obtained an excellent agreement, with our model describing 93% of all experimental variance (Fig. 1e).

### *C. elegans* responds to an effective bacterial density.
The sigmoids shown in Fig. 1d look remarkably similar, their biggest difference being a horizontal shift, which is controlled by the

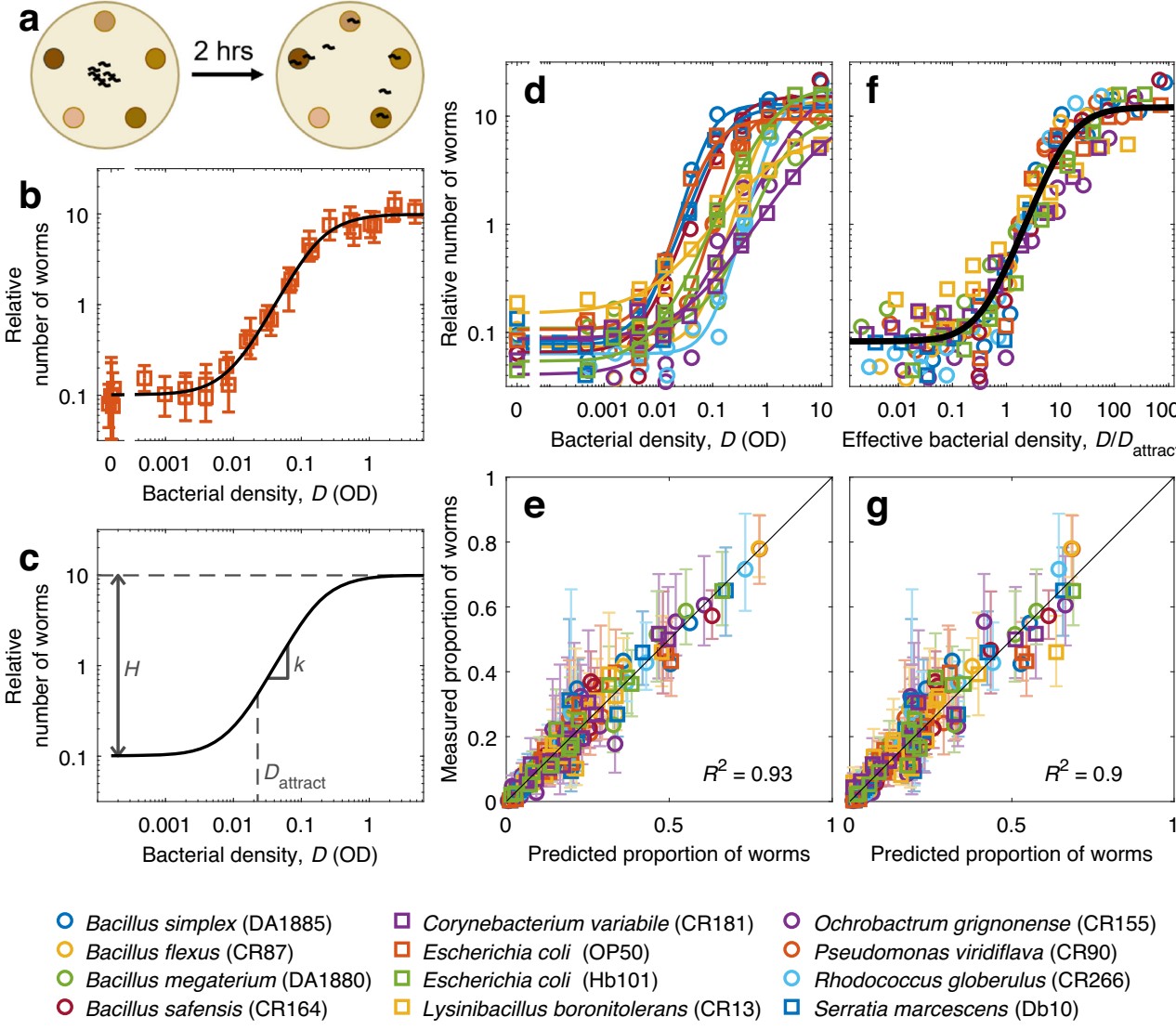

**Fig. 1 C. elegans' response to bacterial density follows a universal sigmoidal trend. a** Experimental scheme: Worms are placed at the center of a regular pentagon formed by 5 food patches of different densities. After 2 h of exploration, worms located at each food patch are counted. **b** Relative number of worms found at each food patch, as a function of bacterial density in the food patch ($D$). Squares: Experimental data for *E. coli* OP50. Line: Fitted sigmoid, following Eq. 1 in Methods. **c** Sigmoid parameters: $H$ is the ratio between the number of worms at the high and low density extremes, $k$ is the slope at the sigmoid's midpoint, and $D_{attract}$ is the density at which the number of worms reaches 5-fold the low-density baseline. **d** Same as (**b**), but for all bacterial strains and without errorbars (see Supplementary Fig. 4 for separate plots for each strain and Supplementary Fig. 5 for the parameters of all sigmoids) **e** Measured proportion of worms in each food patch, versus proportion predicted by the sigmoid, fitted to each strains (Eqs. 1 and 2). **f** Relative number of worms found at each food patch, as a function of effective density ($D/D_{attract}$). Black line: Sigmoid with $H = 146$, $k = 1.4$. **g** Same as (**e**), but with predictions made using effective density and the same sigmoid for all strains. All errorbars show the 95% confidence interval, computed via bootstrapping; see Supplementary Table 1 for sample sizes.

attraction density ($D_{attract}$). Indeed, we found that differences across bacterial strains in parameters $H$ and $k$ are small, while $D_{attract}$ differs up to 50-fold across strains (Supplementary Fig. 5).

We hypothesized that *C. elegans* might address the differences across bacterial strains simply by computing an effective density and reacting to it. We defined effective density as $D/D_{attract}$, and found that this re-scaling of bacterial density removed most of the differences across bacterial strains (Fig. 1f). We therefore re-fitted all our data with a single sigmoid, to describe all strains with the same parameters ($H = 146$, $k = 1.4$, black line in Fig. 1f). This common sigmoid describes all our data almost as accurately as the separate fits (Fig. 1g).

**Attraction to a food patch correlates with its impact on fitness.** We then asked whether *C. elegans*' response was well adapted to choose the food patches that maximize its fitness. To estimate how a period of exploitation of a given food patch impacts *C. elegans*' fitness, we counted the number of eggs laid by an individual worm that feeds on a food patch for 5 h. In order to remove the effect of food preference as much as possible, we encircled the food patch with a copper ring that prevented the worm from escaping, forcing it to stay on the food patch regardless of its preference (Fig. 2a). We found that even in the absence of food, worms lay an average of 3 eggs, which were probably produced while the worms were feeding on high-density OP50 before the start of the assay. The number of eggs increased

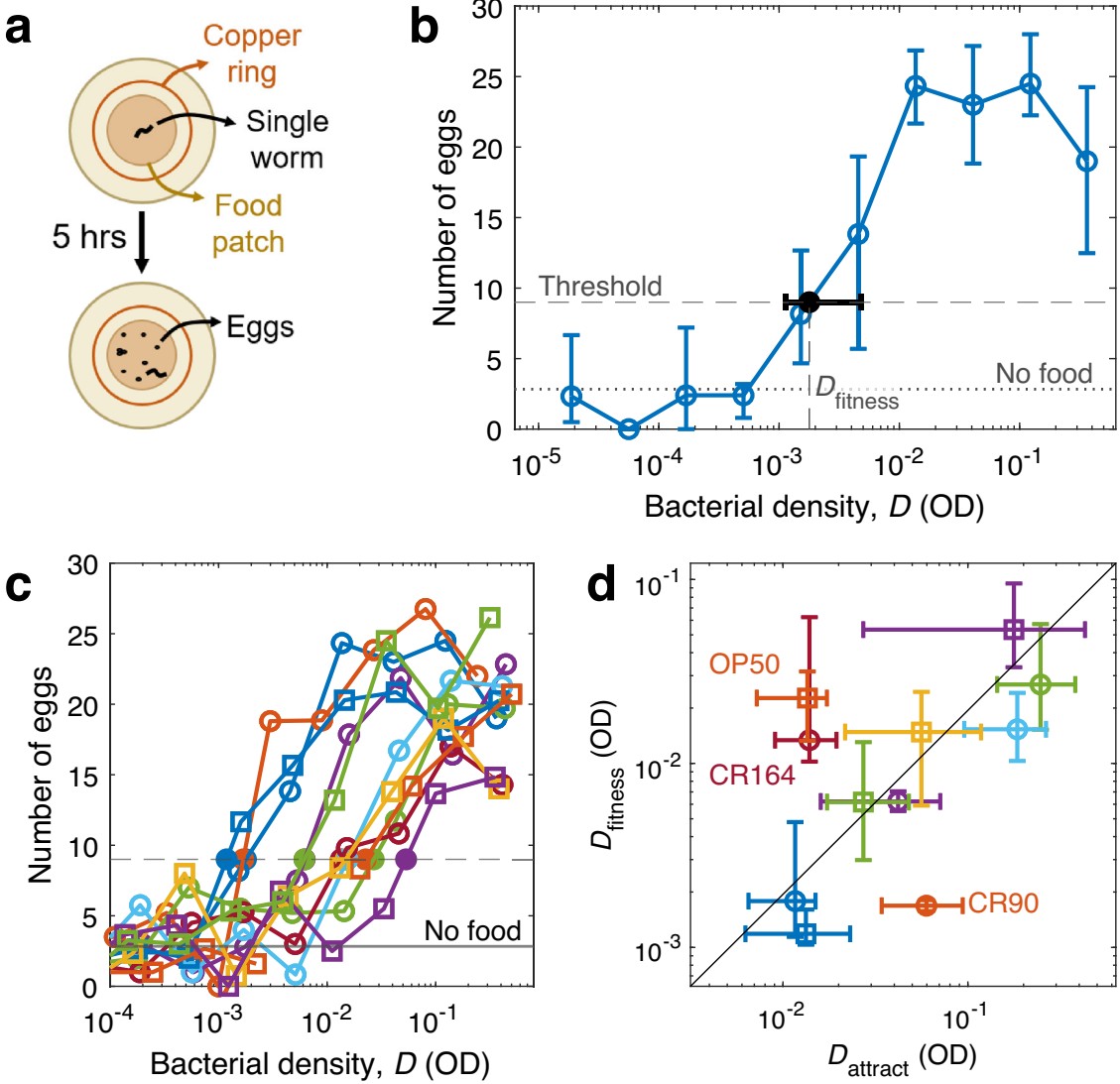

**Fig. 2 Attraction to each bacterial strain correlates with its impact on fitness. a** Experimental scheme to estimate impact on fitness: A single worm was placed in the middle of a food patch, and surrounded by a copper ring to prevent it from escaping. Five hours later, the number of eggs was counted. **b** Number of eggs laid after 5 h on a food patch of DA1885, as a function of the density of the food patch (open circles). Dotted horizontal line: average number of eggs when no food was present. Dashed horizontal line: Threshold chosen to determine when the number of eggs increases with respect to the no-food baseline. Solid dot: Point at which the number of eggs crosses the threshold, which defines $D_{fitness}$. **c** Same as (**b**), but for 11 bacterial strains and without errorbars. **d** Density at which worms start laying more eggs ($D_{fitness}$), versus density at which worms start being attracted to a food patch ($D_{attract}$). Line indicates perfect proportionality between the two variables ($D_{fitness} = 0.2D_{attract}$; the value of the proportionality constant has little consequence, since it depends on the thresholds chosen to define $D_{attract}$ and $D_{fitness}$). Color and shape of all markers identify bacterial strain, following the legend in Fig. 1. All errorbars show 95% confidence intervals, calculated via bootstrapping; see Supplementary Table 1 for sample sizes and Supplementary Data 1 for the data and computer code that generate this figure.

beyond this baseline in the presence of food, reaching an average of 20 eggs at high food densities. This high number of eggs requires active egg production during the assay, since well-fed young adult hermaphrodites usually store 10-15 eggs in their uterus, and only a fraction of them are mature[39]. The number of eggs increased sharply at a given food density that we called $D_{fitness}$, and stabilized at higher densities (Fig. 2b).

Egg-laying increases with bacterial density following a similar trend for all strains, the main difference being a shift in the density at which the increase takes place ($D_{fitness}$) (Fig. 2c). Therefore, an optimal behavioral response maximizing egg production should also follow the same trend for all strains with a shift in density, which is what we found for patch occupancy (Fig. 1). The question is whether the density shifts in food

preference (characterized by the attraction density $D_{attract}$) correspond to those found in egg-laying (characterized by $D_{fitness}$). We found this to be the case: $D_{fitness}$ and $D_{attract}$ are proportional ($p = 0.03$, linear regression), which in a log-log plot corresponds to a line with slope 1 (black line in Fig. 2d).

Three outliers deviate from the general trend (Fig. 2d). One of these outliers is *E. coli* OP50, which was also used to feed the worms before the experiment. This previous experience might explain the deviation, because *C. elegans* can learn odors and tastes associated with beneficial food[20,22], and this learning might increase the attractiveness of OP50 with respect to unfamiliar strains. The other two outliers (*Bacillus safensis* CR164 and *Pseudomonas viridiflava* CR90) cannot be explained in this way. These deviations suggest that *C. elegans'* behavior is near-optimal but not perfectly optimal,

although we must keep in mind that we only measure a proxy for fitness (number of eggs laid in 5 h), and a more accurate measurement might partially explain the outliers. In any case, we conclude that *C. elegans* is using a rule of thumb, focusing on cues that allow it to adapt its behavior to most strains, and probably neglecting others that would be relevant for the outliers.

**Biomass content does not drive food choice**. We next asked what properties of the food determine the observed rule of thumb. We first hypothesized that worms would choose the food patches with the highest biomass density, since biomass density determines the actual amount of food available. So far we have reported bacterial density using optical density (OD), which measures the amount of light absorbed by a bacterial culture. OD is proportional to biomass density for a given bacterial strain, but different bacterial strains have different cell size, shape and composition, which affect light transmission through the culture. Therefore, cultures of different strains at the same OD may have different biomass density. We hypothesized that the different values of attraction density ($D_{attract}$) in terms of OD might in fact reflect the same density threshold in terms of biomass.

To measure biomass density of each bacterial strain, we determined the relation between biomass content and OD. We did this by measuring the weight of dry biomass left after evaporating all the water contained in bacterial cultures of all our strains. If biomass differences were to explain our results, we should find an inverse relationship between biomass content and $D_{attract}$, because strains with twice as much biomass at OD = 1 should have an effective density twice as large, and therefore their attraction density ($D_{attract}$) should be halved.

While we did find a slight negative correlation ($p = 0.05$, linear regression) between biomass content and $D_{attract}$, this correlation is far too weak to explain the differences across bacterial strains: Biomass density changes by less than 3-fold across different strains, while $D_{attract}$ changes up to 50-fold (Fig. 3a; if differences in biomass density were to explain differences in $D_{attract}$, the datapoints should follow the black diagonal, which has slope -1 in this log-log plot). Indeed, using biomass density instead of OD does not improve the predictions of patch occupancy (Supplementary Fig. 6). Therefore, biomass density is not the main driver of food choice in *C. elegans*.

**Cell density drives food choice**. Next, we hypothesized that cell density might be driving the preference. For the same reasons discussed in the previous section, different bacterial strains will have different cell density (i.e. number of cells per unit volume) at the same OD. We determined the cell density in our cultures by a combination of plating and microscopic observations (see Methods). As for biomass, we expected to find an inverse relation between $D_{attract}$ and cell density at OD = 1.

We found an excellent inverse correlation between $D_{attract}$ and cell density at OD = 1 ($p = 0.002$, linear regression), and in this case the correlation was strong enough to explain all the variability in $D_{attract}$ (Fig. 3b). This result indicates that the effective density that drives *C. elegans* behavior is simply bacterial density, but measured in number of cells per unit of volume (or number of cells per unit of surface, once the bacteria are placed on the surface of the agar plate). We confirmed this fact by plotting our original data with the bacterial density measured in cells/mm², and comparing them with a single sigmoid (Fig. 3c).

Therefore, a rule that characterizes a food patch exclusively by its density in cells/mm² (making no distinction across bacterial strains), describes the experimental results with high accuracy

(Fig. 3d). However, we do find a slight drop in accuracy: Our previous model explained 90% of all experimental variance ($R^2 = 0.9$, Fig. 1g), while the current one explains 80% of it ($R^2 = 0.8$, Fig. 3d). This drop in accuracy is probably due to a combination of two issues: First, other factors besides bacterial cell density may contribute to the effective density that drives *C. elegans*' behavior, so this drop in accuracy may reflect actual biological complexity. Second, our current model has substituted the $D_{attract}$ that were fit to our behavioral results for an independent measurement of the cell density of each bacterial strain, and the experimental inaccuracies of this separate measurement must necessarily reduce the accuracy of the overall fit.

In any case, our results robustly indicate that at least 90 % of *C. elegans*' response to bacteria is driven by an effective density (Fig. 1g), and at least 80 % of the response can be explained by a single environmental variable (bacterial cell density, Fig. 3d), which is the most informative one in terms of fitness benefit (as compared to alternatives, such as biomass density).

**Prediction of patch occupancy in mixed environments**. All previous results correspond to experiments in which worms were exposed to a single bacterial strain at a time. We next tested if our rule could predict patch occupancy in environments containing food patches of several different species. To test this, we used *Escherichia coli* OP50 plus two other strains, DA1880 and CR266, which we chose with the sole criterion of having values of $D_{attract}$ (as reported in Supplementary Fig. 5c) covering a wide range. We then performed experiments with five patches of different species and at different densities (Fig. 4a), recorded the number of worms at each food patch after 2 h, and compared these results with the results predicted by our model (using the same parameters as in Fig. 1f). We performed this experiment for many different combinations of densities of the three bacterial strains (Supplementary Fig. 7), obtaining a good overall agreement between predictions and experimental results (Fig. 4b, c).

**Prediction of patch occupancy in mixed patches**. Our results indicate that *C. elegans* simply measures the number of bacteria encountered per unit surface when deciding whether to keep exploiting a food patch or to leave it. This result produces an interesting prediction, which in turn provides a stronger test of our hypotheses. Let's consider a food patch in which two bacterial strains are well mixed. A worm exploring this food patch will be simultaneously exposed to both strains, so any factors such as different cell composition, different cell size or different metabolites, will be perceived near-simultaneously. We don't know how these stimuli combine in *C. elegans*' nervous system, so if they play an important role it would be hard to predict *C. elegans*' response to a mixed food patch. As a reference we define a reasonable null model, assuming that the response to a mixed food patch would be an average of the responses to the corresponding pure patches, weighted by their relative proportions in the mixture. Therefore, we define our null model as any result between the weighted arithmetic and geometric means of the responses to each strain separately (Fig. 5a).

In contrast, if *C. elegans*' response is determined by the cell density of the food patch, we can predict the response to any mixed patch, by computing its effective density (which is the average of both strains' effective densities, because cell density is additive), and use our sigmoidal model to predict the worm's response to it (Fig. 5b). By choosing pairs of bacterial strains with very different $D_{attract}$, we can have cases in which this predicted response is very different from the null model (compare Fig. 5a and Fig. 5b).

To test these predictions, we performed experiments with food patches containing mixtures of CR266, which has a high $D_{attract}$, and DA1885, which has a low $D_{attract}$. We prepared cultures at

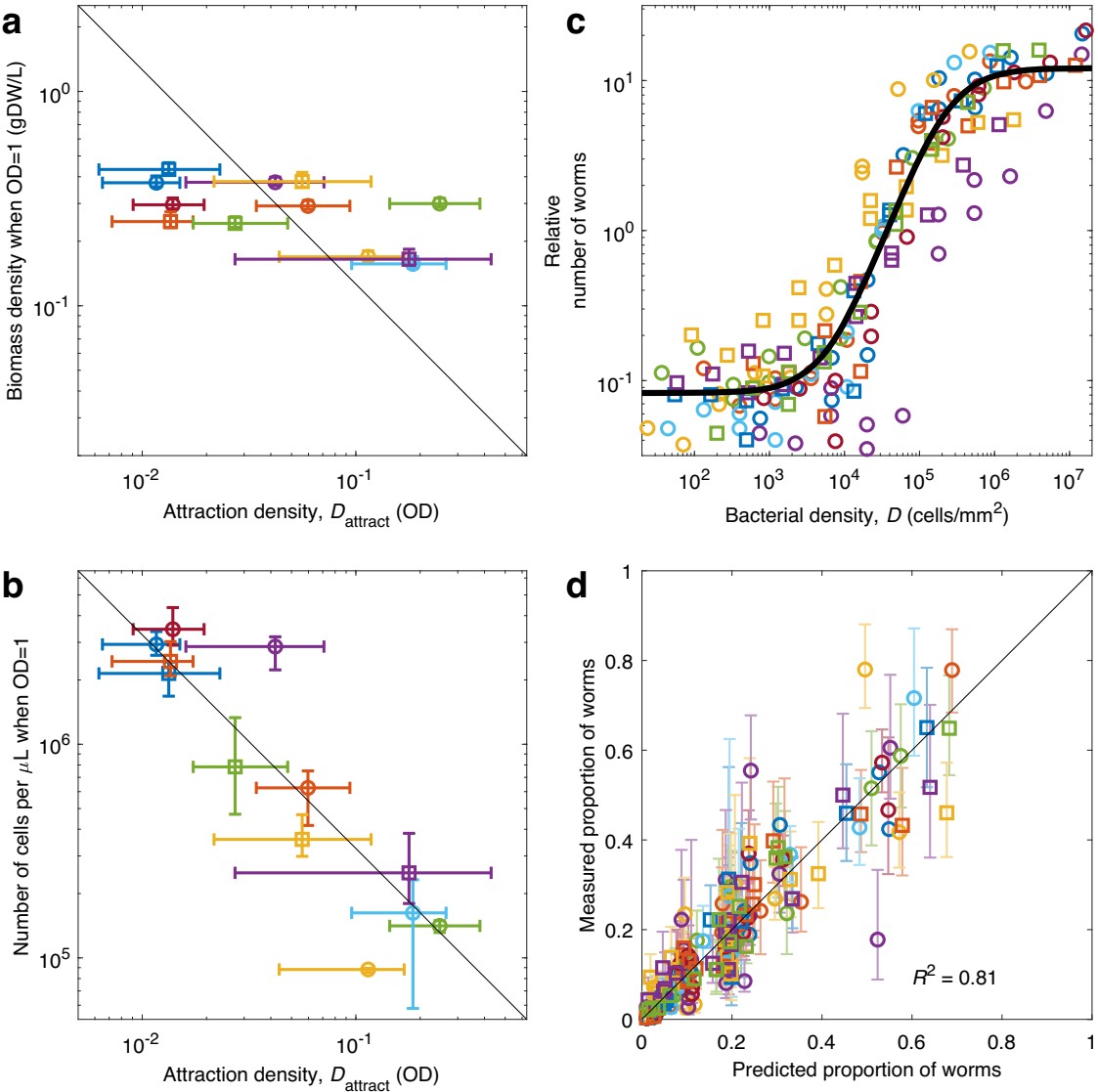

**Fig. 3 C. elegans' rule of thumb is driven by the number of bacteria per unit surface.** Color and shape of markers identify bacterial strains (see legend in Fig. 1). **a** Biomass density for each strain at OD = 1 (measured in grams of dry weight, gDW, per liter), versus its attraction density ($D_{attract}$). Black line: Inverse relation (proportional to $1/D_{attract}$), which would indicate that biomass density is responsible for the observed differences in $D_{attract}$. **b** Number of cells per microliter at OD = 1 for each strain, versus $D_{attract}$ for each strain. Black line: Inverse relation (proportional to $1/D_{attract}$), which indicates that the number of cells is responsible for the observed differences in $D_{attract}$. **c** Relative number of worms found at each food patch, as a function of bacterial density (measured in cells/mm²) in the food patch. Black line: Sigmoid, fitted to all strains. **d** Measured proportion of worms in each food patch, versus proportion predicted by the sigmoid in (**c**). All errorbars show the 95% confidence interval, computed via bootstrapping; see Supplementary Table 1 for sample sizes and Supplementary Data 1 for the data and computer code that generate this figure.

OD = 0.5 for both strains, and prepared 5 mixtures with ratios DA1885:CR266 of 0:1, 0.125:0.875, 0.25:0.75, 0.5:0.5, and 1:0. Given that both pure cultures had OD = 0.5, all 5 mixtures also had OD = 0.5 (range 0.49–0.51). We then ran our experiment, letting worms choose among 5 food patches made from these 5 mixtures (Fig. 5c).

The experimental results follow the predictions of the sigmoidal model, and clearly reject the null model (Fig. 5d). By construction, our null model matches the experimental data exactly for the two single-species patches (fractions 0 and 1), but it falls very far from the experimental data for the mixed patches. Besides the pair of strains presented here, we measured another three pairs (a total of four). Three out of the four pairs showed excellent agreement with our predictions, and in all cases we found better agreement with our predictions than with the null model (Supplementary Fig. 8).

## Discussion

In our experiments, *C. elegans'* response to food across bacterial species was driven by a single variable: Effective food density. This effective density seems to correspond to bacterial cell density (i.e. number of cells per unit surface), with other factors that depend on bacterial strain, such as biomass content, having smaller effects.

Our experimental results may seem to contradict previous studies, but are actually consistent with them. Previous studies[34,35] showed large differences in preferences between certain strains, such as *E. coli* Hb101 and *E. coli* DA837, which is very similar to OP50 and elicits the same behavioral response (Supplementary Fig. 9). In contrast, we found only moderate differences between Hb101 and OP50. But studies showing larger differences were performed on nutrient agar plates, on which bacteria could grow after being deposited on the plate. OP50 is a

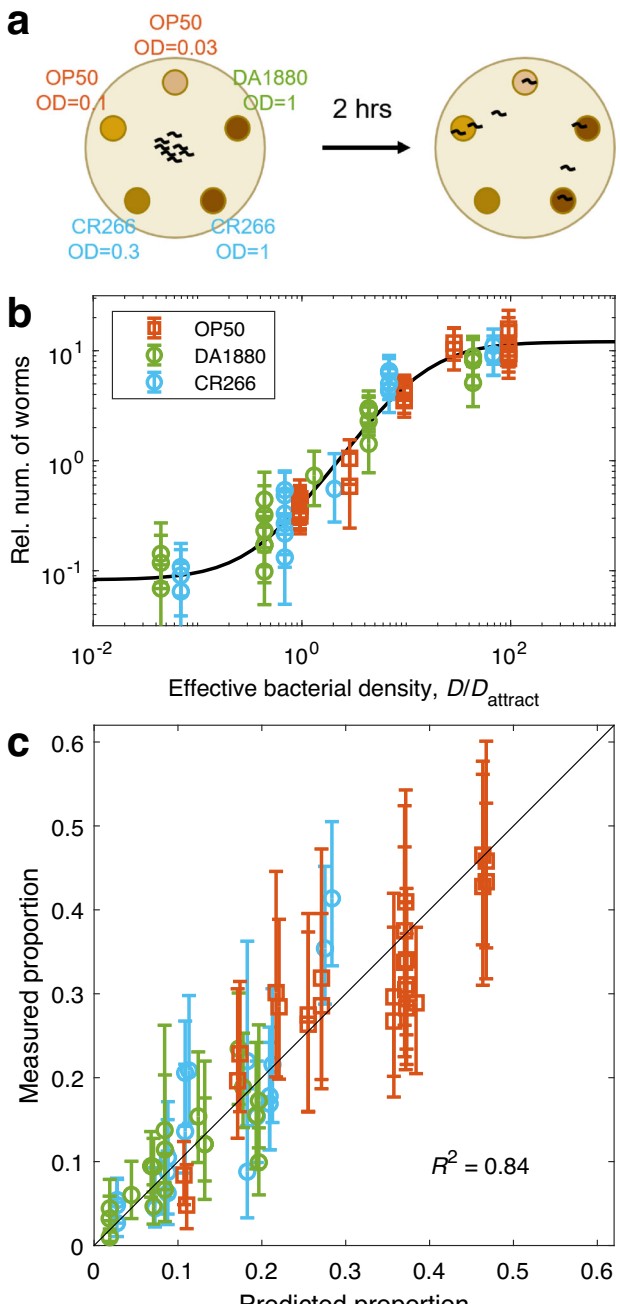

**Fig. 4 Results in environments with food patches of different bacterial species. a** Schematic of the experimental set-up: We placed 5 food patches of different densities and different bacterial species, and counted the number of worms per patch after 2 h. **b** Relative number of worms found at each food patch, as a function of effective density ($D/D_{attract}$). Black line: Sigmoid with $H = 146$, $k = 1.4$. **c** Measured proportion of worms in each food patch, versus proportion predicted by the sigmoid using effective density and the same parameters as in box b. See Supplementary Fig. 7 for results separated by condition, Supplementary Table 1 for sample sizes and Supplementary Data 1 for the data and computer code that generate this figure. All errorbars are 95% confidence intervals, computed via bootstrapping.

uracil auxotroph, and this fact limits its growth on solid media, while Hb101 does not have this limitation and grows to higher densities on agar plates. Therefore, differences across strains observed in previous studies may be attributed to differences in bacterial density. Another apparent contradiction is the evidence

that less-preferred bacteria are hard to eat for *C. elegans*[34,40]. We would not expect that increasing the density of hard-to-eat bacteria would make them as profitable as easier-to-eat strains, so the regularity of our results challenges this previous finding. However, the main reason why some bacterial strains were hypothesized to be harder to eat was that they had a larger cell size[34,40], and strains with larger cell sizes tend to have smaller cell densities at saturation. Therefore, the differences observed in previous studies were also correlated with cell density. A final apparent contradiction comes from studies that have shown that *S. marcescens* (Db10) is a pathogen of *C. elegans*, and actively avoided by the worms[41,42]. We did not find such avoidance behavior, but both the strong pathogenicity and the avoidance response require active production of a toxin by the bacteria, which was not possible in our experimental conditions due to the lack of nutrients in the plates. As a control, we checked that we could reproduce *C. elegans'* avoidance of *S. marcescens* when performing experiments on NGM plates rather than on our experimental plates (Supplementary Fig. 10). In sum, revealing *C. elegans'* foraging rule of thumb required accurate control of bacterial density and decoupling the effect of toxins and other metabolites.

It remains an open question to determine what mechanisms are responsible for *C. elegans'* response to bacterial cell number. A possible hypothesis would be that bacterial number is the easiest way for *C. elegans* to estimate bacterial density. *C. elegans* seems to estimate food availability from the amount of food ingested through its grinder per unit time[31,43], and this measurement might be driven by the number of discrete bodies swallowed. However, this hypothesis would not explain why *C. elegans* lays more eggs when feeding on high cell density food patches, regardless of their biomass density. Two alternative hypotheses would be consistent with this result: First, it might be that certain key nutrients are produced by the bacteria on a cell-by-cell basis, and that both *C. elegans'* attraction to food and its egg-laying capability correlate with their concentration. Second, it might be that high cell number makes it easier for *C. elegans* to find and consume bacteria. This might be especially important at low bacterial densities: At a density of, for example, $10^4$ cells/mm², only about 1% of the surface of a food patch is actually covered by bacteria. Therefore, at this density worms must find individual cells or small clusters of cells, which are sparsely distributed. The success in this search might be the key factor driving *C. elegans'* feeding rate, and it would depend on cell number rather than on total biomass density. We currently have insufficient evidence to decide between these two hypotheses.

The role of oxygen concentration deserves a separate comment, since our results differ from most of previous studies in this respect. *C. elegans* is attracted to low oxygen concentrations, possibly as a means to find food (since dense bacterial populations deplete the oxygen in their immediate vicinity)[32,44–46]. Most previous studies of *C. elegans* foraging used bacterial patches growing on rich media (usually NGM[21,24,26,29,34,44–46] or low-peptone NGM[14,32], where bacteria can still grow), where bacterial metabolism is active, and bacteria are therefore actively depleting oxygen. In contrast, our experiments took place on plates lacking nutrients for the bacteria, and with small amounts of bacteriostatic antibiotics to ensure that bacterial density remained constant during the 24 h that elapsed between the deposition of the bacterial drops and the behavioral experiment. For this reason, we do not expect that our food patches depleted oxygen significantly (at least, oxygen depletion should be much weaker than in other studies). On the one hand, this difference raises the question of whether our results would still hold in situations with active oxygen depletion, a question that we could not answer experimentally because it is hard to control accurately the density of

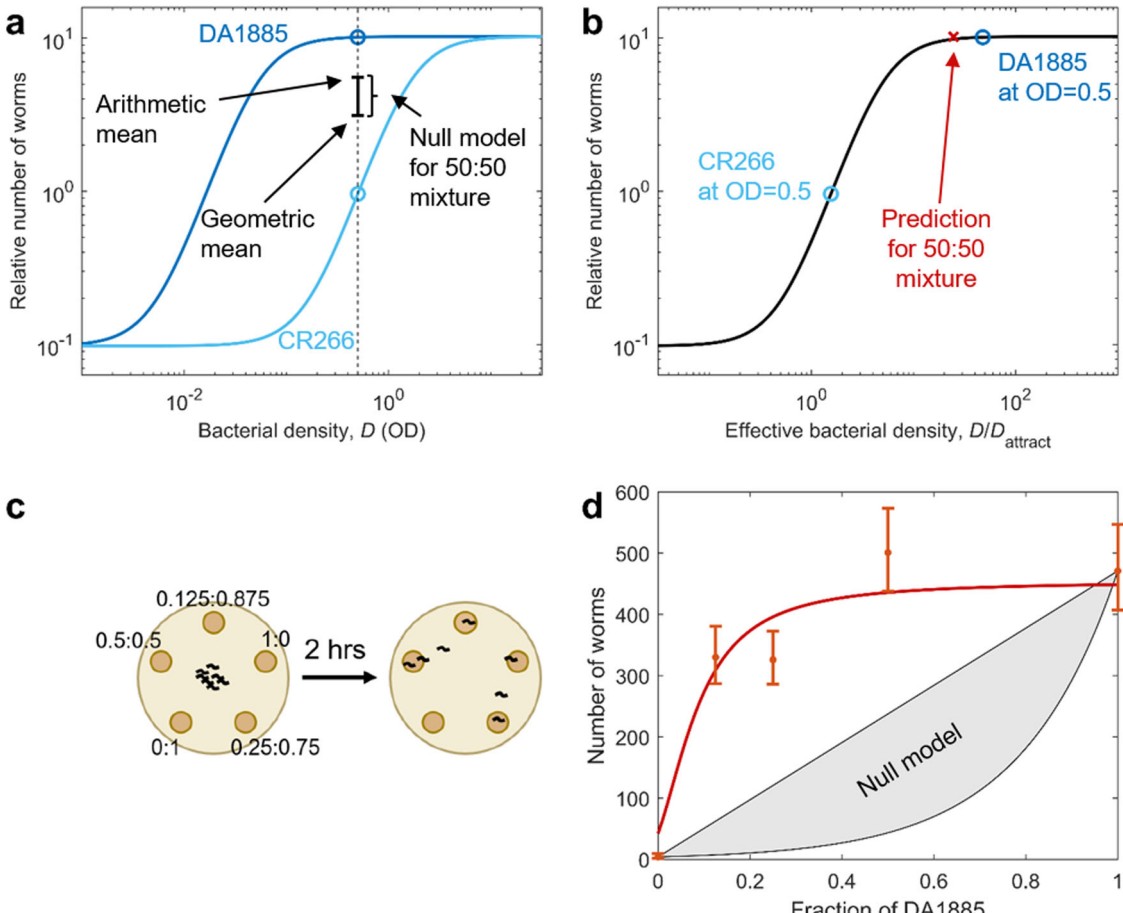

**Fig. 5 The response to mixed food patches shows that effective density is additive. a** Relative number of worms predicted by the sigmoidal model for patches of DA1885 (dark blue) and CR266 (light blue), as a function of bacterial density (measured in OD). Our experiment took place at OD = 0.5 (black dashed line), where DA1885 is about 10 times more attractive than CR266 (circles). The null model for a 50:50 mixture of both strains goes from the geometric mean to the arithmetic mean of the two pure patches (black errorbar). **b** Relative number of worms in a food patch, as a function of its effective bacterial density ($D/D_{attract}$). Circles: Effective density for CR266 and DA1885 at OD = 0.5. Red cross: Effective bacterial density for a 50:50 mixture of CR266 and DA1885 at OD = 0.5 (the effective density of the 50:50 mixture is the arithmetic mean of both effective densities, which in a logarithmic scale is located closer to the highest one). **c** Experimental scheme: Worms were exposed to 5 food patches with different fractions of CR266 and 1885, always at OD = 0.5. Worms at each patch were counted after 2 h. **d** Number of worms at each food patch, as a function of the fraction of DA1885 in the food patch. Dots: Experimental results (errorbars show the 95% confidence interval, computed via bootstrapping). Red line: Prediction from the sigmoidal model. Gray patch: Prediction of the null model. See Supplementary Table 1 for sample sizes and Supplementary Data 1 for the data and computer code that generate this figure.

metabolically active bacteria. On the other hand, our results show that *C. elegans* is capable of foraging efficiently even without the aid of oxygen cues.

A limitation of our study is that we measured a single experimental outcome (patch occupancy). Differences in this outcome emerge from changes in elementary behavioral parameters, such as speed, turning rate, probability of different behavioral states (such as roaming, dwelling and quiescence), etc[24–26,30]. A more detailed study of these parameters might reveal differences that were not apparent here. This higher degree of detail was not possible at the level of throughput and coverage needed to reveal the rule of thumb, but is a natural next step towards unveiling its mechanistic and neural implementation.

A second limitation of our study is that, while our collection of bacterial strains is larger and most diverse than the ones used in most previous studies, we were not able to sample the whole diversity of natural bacteria. In particular, experimental limitations restricted us to bacteria that grow aerobically on LB medium. Also, most of our strains turned out to be rod-shaped (with two of them forming chains or filaments; Supplementary Fig. 11

and Supplementary Table 2), so we don't know whether different shapes would change *C. elegans'* response. However, our strains did show great variability in many other variables: They cover 7 different bacterial families, show more than a 30-fold difference in volume between the smallest and largest strains (Supplementary Table 2), and include both gram-positive and gram-negative bacteria, species with facultative anaerobic growth, differences in nitrate reduction, etc. (Supplementary Tables 3 and 4). This variability suffices to support our main conclusions: The wide diversity in cell size shows that *C. elegans* responds to cell number more strongly than to other ways of measuring bacterial density (such as biomass), and that its response is independent of other characteristics of the bacterial strains (membrane composition, type of metabolism, etc.).

A third limitation of our study is the use of egg-laying as a proxy to fitness. Fitness is an elusive magnitude, and measuring it directly is hard, because it requires long-term measurements over several generations[47,48]. Different proxies of fitness can be used, and previous studies in *C. elegans* used offspring number[49,50], development rate[34,49], number of offspring that reach

adulthood[49,51] longevity[49], and others[52]. While longer-term measurements are more reflective of the actual evolutionary course of the population, they only make sense when the environment remains relevant and stable over a sufficiently long period of time. This was not our case, since our fitness experiments were performed with a single food patch, rather than in a more naturalistic environment. These conditions forced us to use egg-laying as a short-term proxy of fitness, with the additional limitation that we could not wait for eggs to hatch, which might further reduce the accuracy of our measurements (see Methods). However, it is worth noting that the factors affecting long-term fitness in a naturalistic environment (such as future availability of food, future temperature, predation, etc.) are uncertain while an individual is exploiting a food patch. Therefore, foraging decisions are probably driven by the instantaneous benefit of the food on the worm's ability to produce offspring, which is well represented by the number of eggs measured in our experiment.

A fourth limitation of this study is the lack of bacterial growth, which limits the production of bacterial metabolites. These metabolites are probably relevant in natural conditions, as is certainly the case for pathogenic bacteria[41,42]. Our experimental conditions were necessary to properly control bacterial density, and the good correlation of behavior and fitness benefit shows the ecological relevance of our observations. These controlled experimental conditions have revealed a core behavioral mechanism, which produces adaptive response by focusing on the single-most informative environmental variable.

## Methods

**Strains and media**. All experiments were performed with *Caenorhabditis elegans*, strain N2, obtained from the Caenorhabditis Genetics Center (University of Minnesota, https://cgc.umn.edu/), and maintained using standard practices[53]. Worms grew at 22 °C on nematode growth medium (NGM: 3 g/L NaCl, 2.5 g/L peptone, 20 g/L agar, 25 mL/L potassium phosphate buffer pH 6, 1 mM MgSO$_4$, 5 mg/L cholesterol, 1 mM CaCl$_2$), in 100mm-diameter Petri dishes seeded with 200 μL of a saturated culture of *E. coli* OP50 bacteria (grown overnight on LB at 22 °C). The worms were transferred to a fresh dish every 1–3 days to prevent food depletion, so that worms used in any experiment came from a population that had not experienced food depletion for at least 5 generations. To prevent accumulation of mutations, we ensured that our population was never more than 30 generations away from the individuals received from the CGC. We performed all experiments with 48-h old worms, synchronized by bleaching and egg collection (i.e. experiments were performed 48 h after re-feeding the L1 larvae obtained by the bleaching procedure).

*Escherichia coli* (OP50), *Escherichia coli* (Hb101), *Escherichia coli* (DA837), *Serratia marcescens* (Db10), *Bacillus megaterium* (DA1880), and *Bacillus simplex* (DA1885) were obtained from the Caenorhabditis Genetics Center. *Lysinibacillus boronitolerans* (CR13), *Bacillus flexus* (CR87), *Pseudomonas viridiflava* (CR90), *Ochrobactrum grignonense* (CR155), *Bacillus safensis* (CR164), *Corynebacterium variabile* (CR181), *Rhodococcus globerulus* (CR266), *Pseudomonas veronii* (CR220), and *Raoultella terrigena* (CR225) were isolated by us from the gut of *C. elegans* N2 worms who had fed on organic compost (see below).

Bacteria were streaked on NGM plates from a −80 °C glycerol stock, stored at 4 °C, and re-streaked to a fresh plate every 2 weeks to ensure viability. To prepare liquid cultures, we inoculated one or two bacterial colonies in 5 mL of LB medium, and incubated for 24 h, at 22 °C, with orbital shaking at 300 rpm, in a closed 50 mL Falcon tube. Then, 1 μL of this culture was inoculated in either 5 or 10 mL of fresh LB and incubated for another 24 h in the same conditions. *E. coli* Hb101 was an exception, as it took longer than 24 h to reach saturation. In this case we skipped the second inoculation, continuing the incubation of the original culture for a total of 48 h.

**Isolation of *C. elegans* gut bacteria**. The natural microbiota strains of *C. elegans* were isolated by growing *C. elegans* on different types of rotten organic material, followed by washing and sterilizing the worms on the outside, grinding the worms and plating the resulting bacterial suspension on agar plates.

We first prepared heat-killed *E. coli* OP50 by growing OP50 for 24 h in 200 mL tryptic soy broth (Teknova, Hollister, CA, USA) at 37°C, followed by spinning down, resuspending in 4mL S-medium (prepared as described in[53]) and incubation at 80°C for 24 h. This procedure results in 50x *E. coli* OP50 (50x compared to density at saturation).

Two types of food sources were fed to the worms: different types of (i) compost and (ii) rotten fruits and vegetables. Some rotten apples were directly collected from the outside. Other fruits like apples, celery, almonds and parsnip were placed

on local soil from Boston, MA in a household plastic box (Sterilite) with semi-open lid and incubated in the lab at room temperature until the fruits were strongly decayed (~3 weeks). The compost samples were taken from two local compost piles in Boston, MA, that mostly contained kitchen and garden waste. Some amount of phosphate buffered saline and glass beads were added to the samples. The samples were homogenized by vortexing at high speeds. The resulting solution was filtered through a 5 μm filter (Millex-SV 5.0 μm, MerckMillipore, Darmstadt, Germany) to remove bigger particles. The resulting emulsion was spread on S-media agar plates without citrate.

*C. elegans* N2 were first grown on *E. coli* OP50 lawn on NGM plates. The worms were washed off the plates with M9 worm buffer with 0.1% Triton X-100. The worms were let sink down for about 1 min and the supernatant was removed. The worms were resuspended in S-medium containing 100 μg/mL gentamicin and 5x heat-killed OP50 (5x compared to density upon saturation). The worms were incubated in that solution for 24 h at room temperature with gentle shaking (50 mL tube, semi-unscrewed cap). Finally, the worms were washed twice with M9 worm buffer + 0.1% Triton X-100.

The germ-free worms were added to the plates with rotten organic material for around 1 week. After that time the worms were washed off the plates with M9 worm buffer with 0.1% Triton X-100. The worms were washed twice with M9 worm buffer with 0.1% Triton X-100 (centrifugation at 2000g, 10 s). Afterwards the worms were re-suspended in 1 mL ice cold M9 worm buffer with 0.1% Triton X-100 and incubated on ice for 10 mins. 2 μL bleach (Clorox) were added to kill bacteria on the outside of the worm and the worms were incubated for 6 mins on ice. Afterwards the worms were washed 3x with ice cold M9 worm buffer with 0.1% Triton X-100. Single worms were transferred into 0.6 mL reaction tubes (Eppendorf) and ground with a motorized pestle (Kimble Kontes Pellet Pestle, Fisher Scientific) for at least 1 min. The resulting solution was plated onto a tryptic soy broth (Teknova, Hollister, CA, USA) agar plate (2% agar, 150mm-diameter Petri dish). From the resulting colonies, physiologically unique colonies were picked. The colonies were streaked out again on tryptic soy broth agar and checked for contaminations. If contaminations were spotted the bacteria were re-streaked again. Finally, the bacteria were grown in tryptic soy broth at 30°C and stored as glycerol stocks. The species identity was analyzed by 16 S Sanger sequencing (Genewiz, South Plainfield, NJ).

Phylogenetic identity was assigned from 16 S rRNA gene sequence by dada2[54] package trained on green gene 16 S dataset[55]. Strains are available from the authors upon request.

**Preparation of experimental plates**. Assays were run in foraging plates (3 g/L NaCl, 20 g/L agar, 25 mL/L potassium phosphate buffer pH 6, 1 mM MgSO$_4$, 5 mg/L cholesterol, 1 mM CaCl$_2$, 10 mg/L chloramphenicol and 100 mg/L novobiocin). The composition of these plates was designed to prevent bacterial growth, not containing any nutrients for the bacteria, and containing two bacteriostatic antibiotics. We chose this antibiotic cocktail after measuring the Minimum Inhibitory Concentration (MIC) for 6 different bacteriostatic antibiotics and all our bacterial strains. We aimed to prevent bacterial growth while keeping the bacteria as healthy as possible, and we determined that 10 mg/L chloramphenicol and 100 mg/L novobiocin was the best combination to prevent the growth of all strains while keeping the antibiotic concentrations as low as possible. We checked that all bacterial strains remained viable and with constant optical density after 24 h of exposure to this cocktail of antibiotics. Plates were poured 1 week before the assays, and stored at room temperature.

One day before the experiment, bacterial cultures were washed three times with foraging buffer (3 g/L NaCl, 25 mL/L potassium phosphate buffer pH 6, 1 mM MgSO$_4$, 1 mM CaCl$_2$, 10 mg/L chloramphenicol and 100 mg/L novobiocin). After the last wash, we re-suspended the bacteria in foraging buffer, adjusting their OD with a spectrophotometer (Jenway 7200, Cole-Parmer, Staffordshire, UK) to the maximum OD needed for our experiment. We then performed serial dilutions in foraging buffer to obtain all needed densities.

A pipetting robot (OT-2, Opentrons, Long Island City, NY, USA, with custom modifications to handle agar plates) placed drops of bacterial culture on the foraging plates. In all cases, we used 40 μL drops, which spread to a diameter of 11.2 mm on average. Drops were left to dry overnight at 22 °C. Some of our experimental conditions contained a food patch with zero bacterial density. In these cases, we simply left one of the drop positions empty (we did not pipet anything on that position), but then counted the number of worms in that area as if there was a food patch.

**Patch occupancy assays**. We used 55mm-diameter foraging plates with five 40-μL drops of bacteria, forming a regular pentagon with the patch centers 13 mm away from the plate's center. For each experimental condition (consisting of five different food densities), we prepared at least 4 different versions, randomly permuting the position of the 5 densities across the 5 food patches to minimize effects due to relative position of the food patches. We then prepared at least 8 replicates of each version, so we had a total of at least 32 plates per condition. We also randomized the order at which the different conditions were prepared. Food patches were placed on the experimental plates 1 day before the experiment and dried overnight at 22 °C.

48-h old synchronized worms were washed off their cultivation NGM plates with M9 worm buffer + 0.1% Triton X-100 (3 g/L $KH_2PO_4$, 7.52 g/L $Na_2HPO_4.2H_2O$, 5 g/L NaCl, 1 mM $MgSO_4$, 0.1% Triton X-100; Triton X-100 was added to prevent worms from sticking to the pipette tips). To remove all bacteria, we washed the resulting worm suspension 6 times with M9 + 0.1% Triton X-100 using a table-top centrifuge (~5 s spin was enough to pellet the worms while leaving the bacteria in suspension). Worms were then placed in the middle of the experimental plates by the pipetting robot, in drops of 10–15 µL (adjusted for an average of 10 worms per plate). The full wash procedure took between 8 and 10 min and placing the worms took at most 5 min, so at most 15 min elapsed between the breeding plate and the experimental plate.

The worms were left on the plates for 2 h at 22 °C, and then we imaged the plates at 1200 dpi using a scanner (Epson Perfection V850 Pro). Previous protocols that use similar scanners to quantify worm behavior proposed modifications to increase image quality and control temperature[56]. However, we found that unmodified scanners provided good enough image quality for our purposes, and temperature control was not an issue for us because plates were placed on the scanners only briefly at the end of the experiment. Extreme care must be exercised when placing the plates on the scanners, since even a gentle tap may startle the worms and make them leave the food patches. Worms were automatically located in the images using a custom-made program built in Matlab R2019a.

We aimed to have 32 plates for each experimental condition (8 plates for each version with permuted positions), with 10 worms per plate. However, the actual number of plates and worms was lower. First, we removed a small fraction of plates that presented imperfections on the agar surface or non-round food patches. Second, given that we could not control the exact number of worms placed on each experimental plate (just the volume and concentration of worm suspension), we had substantial variability in worm number per plate. To prevent any significant effects from food depletion, we removed from the analysis all plates that had more than 20 worms. After this filtering, we had 29 ± 4 plates per condition and 230 ± 110 worms per condition (mean ± standard deviation). See Supplementary Table 1 for full sample sizes. Then, for each experimental condition, we added up all the worms found at patches of a given density (across all replicates), and added one pseudocount to obtain a less biased estimate[57].

**Fitting and normalization of patch occupancy data**. We assume that the number of worms found in a given food patch is proportional to its attractiveness, $A$, which we define as

$$A = \sqrt{H}\, \frac{1 + 4(D/D_{\text{attract}})^k}{H + 4(D/D_{\text{attract}})^k}, \tag{1}$$

where $H$ is the ratio between its highest and lowest points, $k$ is the slope at the sigmoid's midpoint (in a double-logarithmic plot), and $D_{\text{attract}}$ is the density at which the relative number of worms reaches 5-fold the low-density baseline (Fig. 1c; the low-density baseline is $A(D=0) = 1/\sqrt{H}$, and $A(D = D_{\text{attract}}) = 5\sqrt{H}/(H+4) \approx 5/\sqrt{H}$ when $H \gg 1$).

Then, the proportion of worms present in each food patch in a given plate will be

$$P_i = \frac{A_i}{\sum_{j=1}^{M} A_j} \tag{2}$$

where $P_i$ is the proportion of worms in the $i$-th food patch, $A_i$ is the attractiveness of the $i$-th food patch, and $M$ is the number of patches present in the experiment. Equation 2 is a strong assumption, which holds approximately in our system for reasons that will be explored in a separate article, and which is validated by the excellent goodness of fit of our model (Fig. 1e).

To fit the model's parameters to our experimental data, we maximize the log-likelihood of the model. For one experimental plate, the log-likelihood is

$$\sum_{i=1}^{M} N_i \log(P_i), \tag{3}$$

where $N_i$ is the number of worms found in the $i$-th patch, $M$ is the number of patches (in all our experiments), and $P_i$ is computed using Eqs. 1 and 2. Then, if we have several plates whose results must be described by the same sigmoid, we compute the total log-likelihood as $L_{\text{total}} = \sum_{k=1}^{K} L_k$, where $L_k$ is the log-likelihood of the $k$-th plate (computed with Eq. 3), and $K$ is the number of plates. Note that the conditions on the plates do not need to be identical (for example, each plate may have food patches of different densities).

We then found the set of parameters that maximized $L_{\text{total}}$ using Matlab's 'fmincon' function (Matlab R2019a). Once the optimal parameters are found, Eqs. 1 and 2 provide a good description for the proportion of worms reaching each patch in each separate experiment (Supplementary Fig. 2A).

These two equations are also used to represent all the plots showing predicted vs experimental results (Supplementary Fig. 2b), as presented in Figs. 1e, 1g, 3d of the main text.

In order to show together the experimental data coming from experiments that cover different density ranges for the same bacterial strain (the three columns in Supplementary Fig. 2), we re-normalized the data and computed a relative number of worms, $N_i/N_{\text{ref}}$ (where $N_i$ is the number of worms in the $i$-th food patch, and

$N_{\text{ref}}$ is the number of worms in a reference patch). This normalization would be trivial if we had a reference food patch of some common density in every experiment, but this was not possible in practice: Our range of densities is very wide, and differences in patch occupancy span more than 2 orders of magnitude. When food patches of very dissimilar densities are placed in the same plate, worms accumulate in the high-density ones leaving the low-density ones almost completely empty, and leading to very noisy datasets. For this reason, each individual experiment covered a relatively small range of densities (Supplementary Fig. 2a). Therefore, while neighboring ranges overlap with at least two datapoints, we did not have any one density present in all of them.

To circumvent this issue and be able to represent all the data together, we used our sigmoidal fit to estimate the number of worms we would expect at a virtual reference patch, and used this estimate to re-normalize our experimental data. The procedure works as follows: Consider $C$ different conditions that must be described by the same sigmoid (for example, the three conditions covering different density ranges shown with different colors in Supplementary Fig. 2). Let $M_c$ be the number of patches in the $c$-th condition, and $A_{c,i}$ the attractiveness of the $i$-th patch of the $c$-th condition. Let $N_{c,i}$ the total number of worms on the $i$-th patch of the $c$-th condition (aggregated across all plates with the same condition, as described at the end of section "Patch occupancy assays"). We chose our virtual reference patch to be in the sigmoid's midpoint, and for the $c$-th condition this virtual patch would have the following number of worms:

$$N_{c,\text{ref}} = \frac{\sum_{j=1}^{M_c} N_{c,j}}{\sum_{j=1}^{M_c} A_{c,j}} \tag{4}$$

Then, for each food patch and each condition we plot the relative number of worms ($N_{c,i}/N_{c,\text{ref}}$ for the $i$-th patch of the $c$-th condition). This procedure re-aligns the experimental data from different experiments (Supplementary Fig. 2c), allowing us to present it in a single graph (Supplementary Fig. 2d). However, this alignment depends on the fit itself, and therefore does not provide a reliable visual indication of the goodness of the fit. This visual indication of the goodness of fit is found in the plots showing predicted vs experimental results, which are not affected by this normalization (Figs. 1e, 1g, 3d and Supplementary Fig. 2b).

**Fitness experiments**. We used 35 mm foraging plates with one 40 µL drop of bacteria in the center, placed on the plate the day before the experiment and dried at 22 °C. 48-h old worms were washed from their breeding plates in the same way as for patch occupancy assays. Then, individual worms were fished using a pipette and placed on the bacterial patch (one worm per plate). A copper ring of 2 cm diameter was then lodged into the agar, around the patch, to prevent the worm from escaping. Worms were left on the lawn for 5 h and then put at −20 °C for 5 min. This brief period ensured quick refrigeration of the plates to immobilize the worms and stop egg-laying, without freezing the agar. Then, plates were stored at 4 °C. Worms remained immobile and eggs didn't hatch, so eggs could be counted for at least 2 weeks after the experiment. Eggs were manually counted, excluding any plates where more than one worm was placed by mistake, or where the worm escaped the area delimited by the copper ring. All experiments were performed on the same day to minimize experimental variability, but eggs were counted over the 2 weeks following the experiment. Because of experimental complications, we failed to measure C. elegans' fitness on B. flexus (CR87), so we have measurements for 11 out of our 12 strains. See Supplementary Table 1 for sample sizes.

We counted unhatched eggs (rather than waiting for them to hatch, which would also take into account viability) because of experimental limitations. Our experiment required more than 1500 plates, so it would have been impractical to remove manually every adult after 5 h to allow for more time for the eggs to hatch. Also, the copper ring was not 100% effective at preventing worms from escaping. While it was easy to discard all cases in which the adult had escaped, missing larvae would have decreased the accuracy of our measurements.

Another experimental limitation came from the age of the worms: Our behavioral experiments were performed with 48-h old worms, and fitness experiments had to be performed at the same age. However, at this stage worms are just starting to be fertile, and it may be that some individuals had not yet started to lay eggs at the beginning of our fitness experiment. This factor probably lowers the accuracy of our fitness measurements, but it does not change our conclusions. We randomized thoroughly the order in which worms were added to each experimental condition, so age effects should not produce any systematic bias. Also, all our results have 95% confidence intervals computed via bootstrapping, which take this experimental variability into account.

To determine $D_{\text{fitness}}$, we first found the highest density for which the average number of eggs was below the threshold. Then, we performed linear interpolation between that point and the next one, with bacterial densities in logarithmic scale.

**Determination of bacterial density**. Optical Density (OD) was measured using a spectrophotometer (Jenway 7200, Cole-Parmer, Staffordshire, UK). We found that this spectrophotometer is most accurate for OD's between 0.1 and 1, so we always diluted the bacterial cultures to obtain measurements in this range. Lower OD's could not be measured accurately, so they are inferred from the dilution factors used to prepare them.

To determine density in cells per microliter, we combined plating to determine the amount of colony forming units (CFU) with microscopy to investigate the nature of each CFU.

We determined the CFU density as follows. After determining the OD of the bacterial culture, we performed 10-fold serial dilutions in M9 worm buffer (3 g/L KH$_2$PO$_4$, 7.52 g/L Na$_2$HPO$_4$.2H$_2$O, 5 g/L NaCl, 1 mM MgSO4), and plated four 10-microliter drops of each dilution on an NGM plate. We incubated this plate at room temperature for 48 h, counted the number of colonies in the drops that had around 10 colonies, and used these counts to derive the density of colony-forming units in our original culture. We computed the errorbars by bootstrapping the four drops for each measurement. We performed this plating procedure both before and after washing the bacteria with foraging buffer to prepare our experimental plates (see "Preparation of experimental plates"), and we did not find any consistent differences before and after the wash.

The number of CFU/μL is not identical to the number of cells/μL. First, not all cells that fall on the surface of an agar plate survive and manage to form a colony. To control for this effect, we performed a control using LB plates instead of NGM plates, and we did not find significant differences in viability across these two types of plates, which suggests that viability was high for all strains in both media. Second, each CFU may be a single cell, but it may also be a cluster of cells that clump together and form a single colony. To control for this effect, we studied our bacterial cultures under a microscope (LEICA DM6000). We found that 10 out of our 12 bacterial strains were mostly composed of individual cells, with few clumps, so for these 10 strains CFU/μL is a good estimate of cells/μL. In contrast, *B. megaterium* (DA1880) and *B. flexus* (CR87), form long filaments composed of several cells, and each of these filaments will form a single colony. Using DAPI staining to visualize individual cells in each filament, we counted the number of individual cells per filament, obtaining 9.6 and 8.7 cells per filament on average for *B. megaterium* (DA1880) and *B. flexus* (CR87), respectively. We used these factors to transform the CFU/μL determined from plating to cells/μL for these two strains.

To determine the amount of biomass present in our bacterial cultures, we prepared 200 mL of saturated culture for all strains, washed it 3 times with M9 worm buffer, and resuspended to a volume of 5 mL. We then measured the OD of these suspensions, and placed them in glass tubes that we had previously weighed. We also added three tubes with M9 worm buffer without bacteria, to be able to account for the weight of the salts contained in the buffer. We evaporated all the water by incubating the tubes at 90 C for 24 h, and weighed them again. We checked that longer incubation did not change the weight, meaning that 24 h were enough for all the water to evaporate. We then calculated the biomass contained in each tube by subtracting the weight after incubation minus the weight of the empty tube, and minus the weight corresponding to the salts from the M9 buffer (to calculate this weight we followed the same procedure with tubes that contained only M9 buffer, obtaining a dry weight of 15 g/L, which is close to the theoretical weight we can compute from the recipe of M9 worm buffer). See detailed protocol at dx.https://doi.org/10.17504/protocols.io.kxygxzn44v8j/v1. To compute confidence intervals, we assumed that all weight measurements had the same proportional error observed in the 3 measurements performed to estimate the weight of M9 salts, we estimated the error in OD measurements by performing 10 measurements of the same culture, and we combined these two sources of error using bootstrap.

**Microscopic observations of the bacteria**. Bacteria were cultured from colonies in 5 mL of LB, on a shaker at 300 rpm, at 22.5°C, for 20-28 h. They were then washed in M9 + 0.5% Triton, incubated for 5-10 min in DAPI (1:2 to 1:10 dilution in M9 from 1 mg/mL stock), washed once and then resuspended in M9. One microliter from each culture was then imaged on Agarose 1% pads using a LEICA DM6000 optical microscope with x40 to x100 magnification (see full protocol at dx.https://doi.org/10.17504/protocols.io.n92ldpjr8l5b/v1).

**Statistics and reproducibility**. We computed all errorbars using bootstrap[58]: For a given experimental condition for which we have $P$ replicates (i.e. $P$ experimental plates), we chose $P$ of these replicates randomly, with replacement (so some replicates can be chosen several times, and some will not be chosen). By doing this with all of our experimental conditions, we obtained a bootstrapped dataset. We thus generated at least 1000 bootstrapped datasets. These bootstrapped datasets are an estimate of what we should expect if we repeated our whole experimental process 1000 times, so they give an estimate of the reproducibility of our results[58]. For each errorbar shown in the paper, we computed the corresponding quantity for each of the bootstrapped datasets, removed the most extreme 2.5% of values at each side, and reported the remaining interval as an errorbar. For example, errorbars in $D_{attract}$ were computed fitting our sigmoid to each of the bootstrapped datasets removing the most extreme 2.5% values of $D_{attract}$ resulting from these fits, and reporting the remaining interval.

Significance of correlations was evaluated by fitting linear regression models, using Matlab's fitlm function (Matlab 2019a).

Reproducibility of the results across measurements taken in different laboratories is shown in Supplementary Fig. 4.

**Models for mixed food patches**. Consider mixtures of two strains, A and B, and let $N_A$, $N_B$ be the number of worms found experimentally in the two patches with

pure bacterial cultures. For any other mixture, it computes the weighted arithmetic mean as $P_A N_A + (1 - P_A)N_B$, and the weighted geometric mean as $N_A^{P_A} N_B^{(1-P_A)}$, where $P_A$ is the proportion of strain A in the mixture.

The model based on the sigmoidal rule assumes that the response to both strains follows the same sigmoid, with $H = 146$, $k = 1.4$ and with different attraction densities for each strain, $D_{attract,A}$ and $D_{attract,B}$. Therefore, if $D_A$ and $D_B$ are the optical densities of both pure strains (in all our experiments $D_A = D_B = 0.5$), their effective densities are $D_A/D_{attract,A}$ and $D_B/D_{attract,B}$. The effective density of a mixture with a proportion $P_A$ of strain A and $(1 - P_A)$ of strain B will be $P_A D_A/D_{attract,A} + (1 - P_A)D_B/D_{attract,B}$. Using this effective density and Eqs. 1 and 2, we find the predicted proportion of worms in each food patch. Multiplying these proportions times the total number of worms provides the number of worms in each food patch.

**Reporting summary**. Further information on research design is available in the Nature Portfolio Reporting Summary linked to this article.

## Data availability

Data used in this study, together with the computer code needed to run all the analyses and generate all the figures (except Supplementary Figs. 1 and 11), can be found in Supplementary Data 1.

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

## Acknowledgements

Some strains were provided by the CGC, which is funded by NIH Office of Research Infrastructure Programs (P40 OD010440). We are grateful to Nic Vega for training on C. elegans techniques, Céline Reyes for training and help with microscopy, Anna Mattout for training and help with DAPI staining, Tommaso Biancalani, Jonathan Friedman and other members of Jeff Gore's lab for insightful discussions, and members of IVEP team at the CRCA for comments on the manuscript. AAA received funding from SEVAB PhD school at Université Paul Sabatier, Toulouse, France. C.R. received funding from the European Research Council (ERC) under the European Union's Horizon 2020 research and innovation programme (grant agreement No 948753), the Deutsche Forschungsgemeinschaft (DFG, German Research Foundation) – 468972576 and Cluster of Excellence EXC 2124 "Controlling Microbes to Fight Infections" (CMFI). JG received funding from the National Institutes of Health (P40 OD010440) and the Schmidt Family Foundation. APE received funding from the Human Frontier Science Program (LT000537/2015), a CNRS Momentum grant, a Fyssen Foundation Research Grant, a Gore Family Foundation start-up grant, and grant ANR-22-CE02-0002 (ForAnInstant) from the Agence Nationale de la Recherche (ANR).

## Author contributions

Conceptualization: GM, JG, APE. Data curation: AAA, GM, APE. Formal analysis: AAA, GM, APE. Funding acquisition: AAA, JG, APE. Investigation: GM, AAA, LG, LML, AGE, VRR, MK, MDB, CR, APE. Methodology: GM, CR, APE. Project administration: JG, APE. Resources: CR, APE. Software: APE. Supervision: GM, MDB, JG, APE. Validation: AAA, APE. Visualization: AAA, APE. Writing – original draft: APE. Writing – review & editing: AAA, GM, CR, JG, APE.

## Competing interests

The authors declare no competing interests.
