## [Peer Review File · Communications Biology]

Reviewers' comments:

Reviewer #1 (Remarks to the Author):

Madirolas et al. use *C. elegans* foraging as a model system to study a behavioral algorithm termed rules of thumb. Current literature focuses on simple behavior patterns that acquire or process complex information, whereas, in this manuscript, the authors focused on understanding dimensionality-reduction for a complex behavior like food choice. Using competition assays, the authors experimented with different species and densities of bacteria (food for *C. elegans*) and found that the nematodes tend to favor high-density food sources as more worms were on the high density (number per area) bacterial patch than low-density ones, regardless of the bacterial species and biomass. The authors also found that the nematodes tend to lay more eggs in high-density bacterial patches and concluded that the high-density bacterial patch favoritism provides higher fitness to the nematodes. The overall goal of the manuscript is interesting and important. However, there are several serious dropbacks that require further investigation or clarification.

A major dropback of the paper was that the manuscript failed to demonstrate the complexity of the food source tested. To be able to conclude that the nematodes use "number of bacteria per area" as a low-dimensional representation of complex high-dimensional information, the authors need to provide evidence that the information is indeed complex and high-dimensional. Though the bacterial strains used in the manuscript come from several bacterial families, the variability of the physiology of the bacteria used in terms of cell size, shape, surface chemistry, moisture, texture, and nutrient content (to name a few) is not available to readers. It is vital to show the variability. Moreover, to illustrate that other dimensions are less relevant, it is important to show that this "rule of thumb" can be applied to bacteria with extreme values in the morphological and physiological features mentioned above; for instance, a direct comparison between a small coccus vs an elongated spiral bacterium. In the title of the manuscript and the main text throughout, the authors attributed the favoritism of the nematode towards high bacterial density as a behavior of foraging. Although it is natural to think so since bacteria are the food source for *C. elegans*, however, recent behavioral and neurological studies have shown that the bacteria alter the oxygen concentration in the agar, *C. elegans* is attracted to a specific oxygen level and hence a dense bacterial lawn, and that altering oxygen sensory neural pathway affects such behavior (Demir et al, *elife* (2020); studies from the Bargmann lab starting from de Bono and Bargmann, *Cell* (1998)). I encourage the authors to investigate the possible contribution of oxygen concentration, or change the wording and include this aspect in the discussion.

I have two major critiques of the experimental designs:

In most of the experiments, the authors were in fact using competition assays, which may have artificially boosted the sensitivity towards higher density patches and has a penalty on the low-density patches. This is especially problematic with *C. elegans* as the nematode has the tendency to interact with each other generating a positive feedback loop and potentially undergoing a transcritical bifurcation to form clusters (Chen and Ferrell, *Nature Communications* (2021)). Even though in the response curve measurement the authors noticed the potential bias, and hence broke down the curves into three segments and measured each piece independently, the breaking down resulted in an unbalanced "environment" for each patch. They have to use complicated correction (the correction method was problematic as well) to stitch the three segments. An alternative and more straightforward method to measure a response curve would be to measure the number of worms on an isolated patch with a fixed area on a fixed-sized plate with a fixed number of worms on plate. To measure the fitness of worms under each condition, the authors use the number of eggs laid as a metric for fitness. The idea is acceptable. However, in the experiments, the authors count the number of eggs only 5 hours after the worms have been placed. At the beginning of the experiments, the worms are adults with developed eggs. The resulting number of eggs does not reflect, or at least not only reflect the impact of the bacterial patch on the egg development, but rather a reflection of how

much accumulative time the worm has spent on the patch. Therefore, the number of eggs laid on the patch should not be regarded as a good metric for fitness. The authors should consider other assays to address the fitness problem.

Aside from the major points, a few details require the authors' attention.

In the abstract, "... we performed for *Caenorhabditis elegans* foraging by covering all combinations of food density (across 4 orders of magnitude) and food type (across 12 bacterial strains)". Food density is a continuous variable that cannot be exhausted. The word "all" should be removed.

Introduction. Paragraph 3. Missing a period mark between "arenas" and "Besides".

Results. Section 1. Paragraph 1. "C. elegans is a bacteriophage". I guess the authors wanted to say "C. elegans is a bacteria-eating nematode". Bacteriophage is a word reserved for viruses infecting bacteria.

Results. Section 1. Paragraph 2. The calculation of the relative number is critical. It should be stated in brief in the main text. The normalization process and the method to piece together the 3 segments were unclear. It would be best to express the transformation in mathematical terms.

Results. Section 1. Paragraph 3. "...our sigmoid describes the experimental data remarkably well". Statistics should be provided for such claims.

Results. Section 2. Paragraph 1. "...the remaining 7 strains are bacteria that we isolated from the gut of *C. elegans*". The claim is technically correct but misleading. The fact that *C. elegans* were fed on environmental bacteria beforehand should be stated.

Results. Section 4. Paragraph 3. Figure 2d. It would be better to highlight *E. coli* OP50, *B. safensis* CR164, and *P. viridiflava* CR90 in the plot.

Results. Section 7. Paragraph 4. Figure 4d. Please explain the axes. Why does the y-axis start from zero for the Null model?

Reviewer #2 (Remarks to the Author):

Review for

"A taste for numbers: *Caenorhabditis elegans* foraging follows a low-dimensional rule of thumb"

The authors show that *C. elegans* follows low-dimensional behavioural rules in a set of complex foraging environments. The authors do so by examining patch occupancy in a five-patch arrangement setting of varying food densities, and identifying a single variable, cell density, that explains ~80% of the variation in the observed patch occupancy. This rule applies across 12 bacterial strains as independent food sources, as well as patches consisting of two mixed bacterial strains. The results support a tantalising theory where optimal foraging may be achieved by dimensionality reduction of complex environmental parameter space, which presumably simplifies the necessary neural computation cost and lead to robust behavioural outcomes following a "rule of thumb". The authors confirm the near-optimality of the strategy by measuring a period of egg-laying activity as a proxy for fitness. This study opens up the exciting future possibility of examining the neural implementation of behavioural rules of thumb, in a genetic model organism where such downstream mechanistic research is accessible. This work should be interesting for researchers in the fields of evolution of behaviour, foraging theory, and neural circuitry design.

Overall, the manuscript is well-written and methodology is clearly explained, but I do have several major comments:

1. Some statements and generalisations that the authors make need more support before they can be made.

a. The authors claim that to demonstrate true dimensionality reduction one needs to exhaustively map many combinations of environmental variables such as “all combinations of food density...and food type” (lines 17-19). However the data that were presented appear to only show foraging on one food type (single bacteria strain, or two mixed bacteria at equal OD) at a time, at different densities, without combining across food and densities (see also 3a below). In fact the way the abstract text is written suggests a reduction of dimensionality from two to one (i.e., only density matters, not food type), but in the text body the authors explain it is in fact a dimensionality reduction from four to one, since the food type includes three dimensions extracted from the sigmoidal curve fitting. I find this confusing, on top of not having really explored exhaustive combinations between these parameters.

b. Since the authors derive the rule of thumb by measuring end-point patch occupancy exclusively, it is then important to substantiate the claim that “patch occupancy to be roughly constant at the end of the (2 hour) experiment” (lines 68-70). No data is shown to support this important statement regarding the steady-state nature of patch occupancy in the authors’ experimental design.

c. There are several general statements that should be better qualified with “under our tested conditions”, example: “*C. elegans* response to all bacterial strains follows a universal one-dimensional trend” (lines 60-61), and “*C. elegans* response to food across bacterial species is driven by a single variable” (line 297-298). Also, the point about bacteria not growing and therefore not producing metabolites/toxins that may affect the natural foraging context (lines 320-321, 329-331) should be made more explicitly earlier, to qualify the statement regarding the lack of difference between bacterial strains in generating differential foraging response, especially if the authors are to make arguments regarding “ecological relevance” (line 332).

d. The authors suggest that their high throughput assay allows them to obtain higher accuracy in behavioural measurements that may otherwise be hidden by behavioural variability (lines 41-42). The authors should then demonstrate such increase in accuracy, especially since variability is an inherent design principle for behaviour, so it is important to acknowledge the inherent variability and distinguish it from the noise that comes from the low throughput nature of most behavioural experiments.

2. Data reporting and availability

a. The figure legends should include explicit n numbers for each condition. The patch foraging experiments only report aggregate replicate numbers in the methods section: up to 32 replicates per condition for patch foraging (lines 445-452). The fitness experiments don’t appear to report n numbers in the figure legend nor in the methods section.

b. The analysis code should be made publicly available. Given the small size of the dataset (~2000 still images), perhaps the full dataset could also be shared.

3. Conceptual

a. The authors show mixed food patch experiments (page 10) using pre-mixed food combination at different densities to demonstrate that their rule of thumb holds. A different experiment that would conceptually strengthen their argument further is to use different bacteria type for the patches, at different densities, in a single foraging experiment. This would really demonstrate the dimensionality reduction by showing that the food type does not matter, since as mentioned before (see 1a above), true bacteria type x density combinations in distinct patches have not yet been explored in this work despite the authors’ claim of exhaustive mapping of these in their abstract.

b. The authors should unpack a bit more the concept of “information bottleneck” (lines 105-107), how this necessarily relates to a small nervous system, what this means in ecological and evolutionary terms, since these concepts are central to their proposed theory for why a rule of thumb may exist and what we may learn from it.

c. How do we reconcile between the small nervous system and the complex foraging behaviour (lines 51-53) that co-exist in *C. elegans*, in the light of the authors’ findings? How do the authors’ finding fit in with the other thoroughly developed existing foraging theory? (line 48) These points should further discussed.

d. This is probably beyond the scope of this paper, but I wonder what happens when 20, instead of 10 worms are used for the patch foraging experiments? Does the rule of thumb still apply, or does the effect of social foraging come into play at some critical threshold?

4. Technical

a. Brood size is typically used as a fitness measurement and the full brood period is over 3-4 days for N2 *C. elegans* worms. The authors measured the number of eggs laid over five hours in Day 2 adults at the height of their typical brood period, and admit that this is a proxy and not a full measurement of fitness. There could be other explanations for why eggs laid during this five hours may not necessarily correlate with the overall brood size/fitness. For example it is conceivable that certain food conditions may make the worms lay more eggs, or lay eggs faster, on Day 2 but have overall fewer eggs over the full brood period. The authors should discuss such possibility, and perhaps support their methodology by providing an intuitive explanation for why the five-hour egg laying activity is still a useful read-out for fitness: perhaps it is a good strategy to lay more eggs on better food because the eggs produced on these food may be better quality, etc.

b. Food preference in a foraging context typically refers to preferential choice of one type of food over another, i.e. different bacteria strains. Such food choice experiments were not the scope of this paper and thus I am not sure it is appropriate to compare food preference between different strains by looking at Dattract values that were obtained for single bacteria strain at one time, at different density conditions. Perhaps the authors could better define or clarify their use of terminology “preference” (line 165) and “food choice” (lines 210-211) in their context for disambiguation. How is exactly “preference” (line 182) read from the plot in Figure 2d? “Neglect” (line 184) and “probably neglecting” (line 189) also reads too strong for alternative explanations.

c. Under this set of experimental design, cell density presented as cells/mm² and cells/unit volume are scalable with each other since the surface area of the patch is roughly the same for their experiments. However, one can imagine a situation where the same volume leads to a different surface area, for example because the surface tension of the hydrogel changes and the spotted liquid food spreads to a bigger patch with a larger surface area. Then would the key variable that worms respond to be cell density as measured by unit surface or by unit volume?

d. Lines 86-87: what does a “control patch without bacteria” look like and how was this control performed?

Minor comments:

1. The authors mentioned some environmental variables such as spatial arrangement (line 35) that were not tested, presumably since the patches are randomly positioned on the vertices of the

pentagon. Given the high n number for the patch foraging experiments (up to 32 replicates per food type), perhaps the effect of spatial arrangement is also possible to explore with the existing dataset?

2. The point of determining whether biomass or cell density is the key factor for determining effective density can be better introduced (line 191). Right now these two possibilities read as if they are additional environmental variables to, rather than alternative explanations for, effective density. Also, these are properties of the food and I would suggest rewording them as such, rather than "sensory cues" (line 191) since the authors do not directly explore sensory input onto the nervous system per se. "Amount of food eaten should be the main driver of foraging behavior" (line 191-192) requires a citation and seems a bit out of place here to be stated a priori.

3. Typos

- a. Line 49: "high offspring" \diamond high offspring number
- b. Line 65: "bacteriophage" \diamond bacterivore
- c. Line 223: "collapse"?
- d. Line 275: "run" \diamond ran
- e. Line 346: "48-old" \diamond "48-hour old"

All line numbers in this document refer to the version with track changes.

Reviewers' comments:

Reviewer #1 (Remarks to the Author):

Madirolas et al. use *C. elegans* foraging as a model system to study a behavioral algorithm termed rules of thumb. Current literature focuses on simple behavior patterns that acquire or process complex information, whereas, in this manuscript, the authors focused on understanding dimensionality-reduction for a complex behavior like food choice. Using competition assays, the authors experimented with different species and densities of bacteria (food for *C. elegans*) and found that the nematodes tend to favor high-density food sources as more worms were on the high density (number per area) bacterial patch than low-density ones, regardless of the bacterial species and biomass. The authors also found that the nematodes tend to lay more eggs in high-density bacterial patches and concluded that the high-density bacterial patch favoritism provides higher fitness to the nematodes. The overall goal of the manuscript is interesting and important. However, there are several serious dropbacks that require further investigation or clarification.

1.1- A major dropback of the paper was that the manuscript failed to demonstrate the complexity of the food source tested. To be able to conclude that the nematodes use “number of bacteria per area” as a low-dimensional representation of complex high-dimensional information, the authors need to provide evidence that the information is indeed complex and high-dimensional. Though the bacterial strains used in the manuscript come from several bacterial families, the variability of the physiology of the bacteria used in terms of cell size, shape, surface chemistry, moisture, texture, and nutrient content (to name a few) is not available to readers. It is vital to show the variability. Moreover, to illustrate that other dimensions are less relevant, it is important to show that this “rule of thumb” can be applied to bacteria with extreme values in the morphological and physiological features mentioned above; for instance, a direct comparison between a small coccus vs an elongated spiral bacterium.

We have performed microscopy imaging to accurately characterize the size and shape of our bacterial strains, finding that they indeed cover a wide range of sizes. In particular, we found a >20-fold difference in volume between the largest and the smallest strain (Figure S10, table S2). We have also included two supplementary tables with information about their characteristics (tables S3 and S4, and supplementary references 1-29).

We did find that most of our bacteria are rod-shaped, although two of them associate to form long filaments. We admit that this low diversity in shape is a limitation of our study, and we have added a paragraph acknowledging this limitation. This paragraph (lines 514-526) reads as follows:

“A second limitation of our study is that, while our collection of bacterial strains is larger and most diverse than the ones used in most previous studies, we were not able to sample the whole diversity of natural bacteria. In particular, experimental limitations restricted us to

bacteria that grow aerobically on LB medium. Also, most of our strains turned out to be rod-shaped (with two of them forming chains or filaments; **Figure S10** and **Table S2**), so we don't know whether different shapes would change *C. elegans*' response. However, our strains did show great variability in many other variables: They cover 7 different bacterial families, show more than a 20-fold difference in volume between the smallest and largest strains (**Table S2**), and include both gram-positive and gram-negative bacteria, species with facultative anaerobic growth, differences in nitrate reduction, etc. (**Tables S3 and S4**). This variability suffices to sustain our main conclusions: The wide diversity in cell size shows that *C. elegans* responds to cell number more strongly than to other ways of measuring bacterial density (such as biomass), and that its response is independent of other characteristics of the bacterial strains (membrane composition, type of metabolism, etc.)."

1.2- In the title of the manuscript and the main text throughout, the authors attributed the favoritism of the nematode towards high bacterial density as a behavior of foraging. Although it is natural to think so since bacteria are the food source for *C. elegans*, however, recent behavioral and neurological studies have shown that the bacteria alter the oxygen concentration in the agar, *C. elegans* is attracted to a specific oxygen level and hence a dense bacterial lawn, and that altering oxygen sensory neural pathway affects such behavior (Demir et al, *elife* (2020); studies from the Bargmann lab starting from de Bono and Bargmann, *Cell* (1998)). I encourage the authors to investigate the possible contribution of oxygen concentration, or change the wording and include this aspect in the discussion.

We find two separate issues here:

About our language throughout the manuscript: We use “foraging” as a general term that encompasses all the mechanisms that *C. elegans* uses to find and consume bacteria. In this context, we understand *C. elegans*' attraction to oxygen as one of these mechanisms, and hence part of its foraging behavior. For this reason, and to avoid excessive language complexity, we think that the wording in most of the manuscript may remain unchanged. While oxygen sensing may serve other functions besides foraging, we believe that most authors agree with the interpretation that its main function may be detecting food (see for example Cheung...de Bono, *Current Biology* 2005, Rogers...de Bono, *Current Biology* 2006, Hums et al. *eLife* 2016, Milward et al. *PNAS* 2011).

About the role of oxygen in our experiments: We agree that this is a very important mechanism, and that the paper was missing a more thorough discussion. We have added a general paragraph about possible mechanisms, and one specific paragraph about oxygen (which in fact is particularly interesting, because oxygen depletion is probably much weaker in our case than in previous studies). These two paragraphs (lines 471-506) read as follows:

“It remains an open question what mechanisms are responsible for *C. elegans*' response to bacterial cell number. A possible hypothesis would be that bacterial number is the easiest way for *C. elegans* to estimate bacterial density. *C. elegans*' seems to estimate food availability from the amount of ingested through its grinder per unit time,^{34,45} and this measurement might be driven by the number of discrete bodies swallowed. However, this hypothesis would not explain why *C. elegans* lays more eggs when feeding on high cell density food patches, regardless of their biomass density. Two alternative hypotheses would be consistent with this result: First, it

might be that a key nutrient is produced by the bacteria on a cell-by-cell basis, and that both *C. elegans* attraction to food and its egg-laying capability correlate with the concentration of this nutrient. Second, it might be that high cell number makes it easier for *C. elegans* to find and consume bacteria. This might be especially important at low bacterial densities: At a density of, for example, 10^4 cells/mm², only about 1% of the surface of a food patch is actually covered by bacteria. Therefore, at this density worms must find individual cells or small clusters of cells, which are sparsely distributed. The success in this search might be the key factor driving *C. elegans*' feeding rate, and it would depend on cell number rather than on total biomass density. We currently have insufficient evidence to decide between these two hypotheses.

The role of oxygen concentration deserves a separate comment, since our results differ from most of previous studies in this respect. *C. elegans*' is attracted to low oxygen concentrations, possibly as a means to find food (since dense bacterial populations deplete the oxygen in their immediate vicinity).^{35,46-48} Most previous studies of *C. elegans* foraging used bacterial patches growing on rich media (usually NGM^{22,25,27,29,32,46-48} or low-peptone NGM^{15,35}, where bacteria can still grow), where bacterial metabolism is active, and bacteria are therefore actively depleting oxygen. In contrast, our experiments took place on plates lacking nutrients for the bacteria, and with small amounts of bacteriostatic antibiotics to ensure that bacterial density remained constant during the 24 hours that elapsed between the deposition of the bacterial drops and the behavioral experiment. For this reason, we do not expect that our food patches depleted oxygen significantly (at least, oxygen depletion should be much weaker than in other studies). On the one hand, this difference raises the question of whether our results would still hold in situations with active oxygen depletion, a question that we could not answer experimentally because it is hard to control accurately the density of metabolically active bacteria. On the other hand, our results show that *C. elegans* is capable of foraging efficiently even without the aid of oxygen cues."

1.3- I have two major critiques of the experimental designs:

In most of the experiments, the authors were in fact using competition assays, which may have artificially boosted the sensitivity towards higher density patches and has a penalty on the low-density patches. This is especially problematic with *C. elegans* as the nematode has the tendency to interact with each other generating a positive feedback loop and potentially undergoing a transcritical bifurcation to form clusters (Chen and Ferrell, Nature Communications (2021)). Even though in the response curve measurement the authors noticed the potential bias, and hence broke down the curves into three segments and measured each piece independently, the breaking down resulted in an unbalanced "environment" for each patch. They have to use complicated correction (the correction method was problematic as well) to stitch the three segments. An alternative and more straightforward method to measure a response curve would be to measure the number of worms on an isolated patch with a fixed area on a fixed-sized plate with a fixed number of worms on plate.

We have added the following controls:

- We have added a validation experiment to confirm that our normalization (or correction) works as expected. This validation includes experiments with two food patches having a greater difference in density (so that they cover the whole range of the sigmoid without

being affected by the normalization), and also experiments with a single food patch, as suggested. This validation is shown in Figure S3 and cited in line 105, and we reproduce it here:

Figure S3: Validation of the normalization procedure. Relative number of worms found at each food patch, as a function of bacterial density in the food patch (D). Squares: Experimental data for *E. coli* OP50; errorbars show the 95% confidence interval, computed via bootstrapping. Line: Fitted sigmoid, following **Equation 1 in Methods**. The validation was performed as follows: We first studied 5 experimental conditions, each of them with 5 food patches, collectively covering the whole density range (orange points). We applied our normalization procedure as described in the methods to these 5 dataset, obtaining the sigmoid (black line). Then, we performed three separate experiments, one of them with only two food patches of densities 0.008 and 2.16 (black squares), and another two experiments with a single food patch, with densities 0.03 and 0.81, in each of the two single-patch experiments (blue squares). We used the same geometry for all experiments, so for these control experiments the densities at the five corners of the pentagon were $[0, 0, 0, 0.008, 2.16]$, $[0, 0, 0, 0, 0.03]$, $[0, 0, 0, 0, 0.81]$. The results from each of these experiments were multiplied times a renormalization constant, to transform from absolute to relative number of worms. Note that this normalization cannot change the ratio between the different datapoints within each experiment. These ratios agreed exactly with the ones predicted by the sigmoid (black and blue squares). See **Supplementary Data 1** for the data and computer code that generate this figure.

- We have added a longer discussion about worm density in our experiments. Most results showing collective effects in *C. elegans*, require high worm densities (0.15 worms/mm in the case of Chen and Ferrell's paper; this would be equivalent to ~500

worms in our experiments). We always worked with less than 20 worms per plate (<0.007 worms/mm on average), to ensure that these effects are negligible in our experiments. This is now discussed in lines 87-90, which read as follows:

“We placed approximately 10 worms per plate, and discarded any plates with more than 20 worms. We kept this low number of individuals per plate to prevent food depletion, and also to prevent collective effects such formation of clusters^{37,38}, or networks³⁹, which require much higher worm densities (hundreds to millions of worms per plate).”

- We have added an extra experiment in which patches of different bacterial species coexisted in the same plate. While this experiment is not a direct response to this comment, it also strengthens the case that the particular combination of food patches in a plate plays little role, with our sigmoid predicting patch occupancy accurately. This experiment is discussed in lines 363-373, the new Figure 4, and Figure S6. We reproduce here lines 363-373 and the new Figure 4:

“The one-dimensional rule predicts patch occupancy in mixed environments

All previous results correspond to experiments in which worms were exposed to a single bacterial strain at a time. We next tested if our one-dimensional rule could predict patch occupancy in environments containing food patches of several different species. To test this, we used *Escherichia coli* OP50 plus two other randomly chosen strains, which we chose randomly just making sure that the three strains covered a wide range of D_{attract} (DA1880 and CR266) . We then performed experiments with five patches of different species and at different densities (**Figure 4a**), recorded the number of worms at each food patch after 2 hours, and compared these results with the results predicted by our one-dimensional model (using the same parameters as in **Figure 1f**). We found good agreement between our predictions and the experimental results (**Figure 4b**). We performed this experiment for many different combinations of densities of the three bacterial strains (**Figure S6**), obtaining a good overall agreement between predictions and experimental results (**Figure 4c**).”

Figure 4: Results in environments with food patches of different bacterial species. **a.** Schematic of the experimental set-up: We placed 5 food patches of different densities and different bacterial species, and counted the number of worms per patch after 2 hours. **b.** Relative number of worms found at each food patch, as a function of effective density ($D/D_{attract}$). Black line: Sigmoid with $H = 146$, $k = 1.4$. **c.** Measured proportion of worms in each food patch, versus proportion predicted by the sigmoid using effective density and the same parameters as in box b. See **Figure S6** for results separated by condition, **Table S1** for sample sizes and **Supplementary Data 1** for the data and computer code that generate this figure. All errorbars are 95% confidence intervals, computed via bootstrapping.

1.4- To measure the fitness of worms under each condition, the authors use the number of eggs laid as a metric for fitness. The idea is acceptable. However, in the experiments, the authors count the number of eggs only 5 hours after the worms have been placed. At the beginning of the experiments, the worms are adults with developed eggs. The resulting number of eggs does not reflect, or at least not only reflect the impact of the bacterial patch on the egg development, but rather a reflection of how much accumulative time the worm has spent on the patch. Therefore, the number of eggs laid on the patch should not be regarded as a good metric for fitness. The authors should consider other assays to address the fitness problem.

We have addressed this issue in the following ways:

We now clarify that worms indeed lay some eggs even in the absence of food, and that we always compare with this baseline. The number of eggs observed at high food densities is sufficiently high to require active production of new eggs during the assay, and we now point this out in the text. These clarifications are in lines 244-250, and read as follows:

“We found that even in the absence of food, worms lay an average of 3 eggs, which were probably produced while the worms were feeding on high-density OP50 before the start of the assay. The number of eggs increased beyond this baseline in the presence of food, reaching an average of 20 eggs at high food densities. This high number of eggs requires active egg production during the assay, since well-fed young adults hermaphrodites usually store 10-15 eggs in their uterus, and only a fraction of them are mature.⁴¹ The number of eggs increased sharply at a given food density that we called D_{fitness} , and stabilized at higher densities (**Figure 2b**).”

Also, we have added a paragraph in Methods explaining why we used number of eggs as a metric, rather than alternatives. This paragraph is in lines 769-774 and it reads as follows:

“We counted unhatched eggs (rather than waiting for them to hatch, which would also take into account viability) because of experimental limitations. Our experiment required more than 1500 plates, so it would have been impractical to remove manually every adult after 5 hours to allow for more time for the eggs to hatch. Also, the copper ring was not 100% effective at preventing worms from escaping. While it was easy to discard all cases in which the adult had escaped, missing larvae would have decreased the accuracy of our measurements.”

Finally, we have added a paragraph in the Discussion, highlighting the limitations of our use of number of eggs as a proxy of fitness impact. This paragraph is in lines 527-543 and reads as follows:

“A third limitation of our study is the use of egg-laying as a proxy to fitness. Fitness is an elusive magnitude, and measuring it directly is hard, because it requires long-term measurements over several generations^{49,50}. Different proxies of fitness can be used, and previous studies in *C. elegans* used offspring number^{51,52}, development rate^{27,51}, number of offspring that reach adulthood^{51,53} longevity⁵¹, and others⁵⁴. While longer-term measurements are more reflective of the actual evolutionary course of the population, they only make sense when the environment remains relevant and stable over a sufficiently long period of time. This was not

our case, since our fitness experiments were performed with a single food patch, rather than in a more naturalistic environment. These conditions forced us to use egg-laying as a short-term proxy of fitness, with the additional limitation that we could not wait for eggs to hatch, which might further reduce the accuracy of our measurements (see Methods). However, it is worth noting that the factors affecting long-term fitness in a naturalistic environment (such as future availability of food, future temperature, predation, etc.) are uncertain while an individual is exploiting a food patch. Therefore, foraging decisions are probably driven by the instantaneous benefit of the food on the worm's ability to produce offspring, which is well represented by the number of eggs measured in our experiment."

1.5- Aside from the major points, a few details require the authors' attention.

In the abstract, "... we performed for *Caenorhabditis elegans* foraging by covering all combinations of food density (across 4 orders of magnitude) and food type (across 12 bacterial strains)". Food density is a continuous variable that cannot be exhausted. The word "all" should be removed.

We have replaced "all" by "systematically" (line 20), so the sentence now reads as follows:

"Experimental proof of a dimensionality reduction requires an exhaustive mapping of all relevant combinations of several environmental parameters, which we performed for *Caenorhabditis elegans* foraging by covering systematically combinations of food density (across 4 orders of magnitude) and food type (across 12 bacterial strains)."

Introduction. Paragraph 3. Missing a period mark between "arenas" and "Besides".

Corrected.

Results. Section 1. Paragraph 1. "C. elegans is a bacteriophage". I guess the authors wanted to say "C. elegans is a bacteria-eating nematode". Bacteriophage is a word reserved for viruses infecting bacteria.

Thank you for noticing this! We have corrected it.

Results. Section 1. Paragraph 2. The calculation of the relative number is critical. It should be stated in brief in the main text. The normalization process and the method to piece together the 3 segments were unclear. It would be best to express the transformation in mathematical terms.

We have extended the description of the normalization in Methods, being more explicit and expressing every step in mathematical terms (lines 693-733).

Due to feedback from colleagues who read different versions of the paper, we feared that a partial discussion of this method in the main text might be confusing (especially for readers without a strong mathematical background). For this reason, we have emphasized in the main text that this normalization is just useful for visualization purposes, given that we estimate the goodness of fit from the "predicted vs experimental" plots, which are not affected by this normalization (Figures 1e, 1g, 3d, 4c).

(See also our response to the following comment). The end of the second paragraph of section 1 (lines 103-105) now reads as follows:

“For visualization purposes, we normalized the number of worms in each experiment with respect to a virtual reference to obtain a relative number of worms comparable across conditions (**Figures S2, S3** and **Methods**).”

Results. Section 1. Paragraph 3. “...our sigmoid describes the experimental data remarkably well”. Statistics should be provided for such claims.

We agree with this comment. We admit that, because of the normalization, it does not make sense to claim good fits in the sigmoidal plots. Therefore, we have removed this claim and all other claims of good fit when discussing the sigmoidal plots (lines 116, 124, 185, 329). The relevant statistics are shown in the predicted vs. real plots, where we report *R* squared.

Results. Section 2. Paragraph 1. “...the remaining 7 strains are bacteria that we isolated from the gut of *C. elegans*”. The claim is technically correct but misleading. The fact that *C. elegans* were fed on environmental bacteria beforehand should be stated.

This sentence now reads (line 122) “the remaining 7 strains are bacteria that we isolated from the gut of *C. elegans* worms grown in the lab on different types of natural compost (see **Methods**).”

Results. Section 4. Paragraph 3. Figure 2d. It would be better to highlight *E. coli* OP50, *B. safensis* CR164, and *P. viridiflava* CR90 in the plot.

Done. This is the new Figure 2:

Results. Section 7. Paragraph 4. Figure 4d. Please explain the axes. Why does the y-axis start from zero for the Null model?

We have corrected the figure caption. For box d, it now reads “(d) Number of worms at each food patch, as a function of the fraction of DA1885 in the food patch. Dots: Experimental results (errorbars show the 95% confidence interval, computed via bootstrapping). Red line: Prediction from the sigmoidal model. Gray patch: Prediction of the null model.”

The null model does not start exactly at zero, but rather at the exact value of the number of worms for fraction 0 of DA1885. We have clarified this by adding the following sentence (line 401): “By construction, our null model matches the experimental data exactly for the two single-species patches (fractions 0 and 1), but it falls very far from the experimental data for the mixed patches.”

Reviewer #2 (Remarks to the Author):

Review for

“A taste for numbers: *Caenorhabditis elegans* foraging follows a low-dimensional rule of thumb”

The authors show that *C. elegans* follows low-dimensional behavioural rules in a set of complex foraging environments. The authors do so by examining patch occupancy in a five-patch arrangement setting of varying food densities, and identifying a single variable, cell density, that explains ~80% of the variation in the observed patch occupancy. This rule applies across 12 bacterial strains as independent food sources, as well as patches consisting of two mixed bacterial strains. The results support a tantalising theory where optimal foraging may be achieved by dimensionality reduction of complex environmental parameter space, which presumably simplifies the necessary neural computation cost and lead to robust behavioural outcomes following a "rule of thumb". The authors confirm the near-optimality of the strategy by measuring a period of egg-laying activity as a proxy for fitness. This study opens up the exciting future possibility of examining the neural implementation of behavioural rules of thumb, in a genetic model organism where such downstream mechanistic research is accessible. This work should be interesting for researchers in the fields of evolution of behaviour, foraging theory, and neural circuitry design.

Overall, the manuscript is well-written and methodology is clearly explained, but I do have several major comments:

2.1. Some statements and generalisations that the authors make need more support before they can be made.

2.1.a. The authors claim that to demonstrate true dimensionality reduction one needs to exhaustively map many combinations of environmental variables such as “all combinations of food density...and food type” (lines 17-19). However the data that were presented appear to only show foraging on one food type (single bacteria strain, or two mixed bacteria at equal OD) at a time, at different densities, without combining across food and densities (see also 3a below). In fact the way the abstract text is written suggests a reduction of dimensionality from two to one (i.e., only density matters, not food type), but in the text body the authors explain it is in fact a dimensionality reduction from four to one, since the food type includes three dimensions extracted from the sigmoidal curve fitting. I find this confusing, on top of not having really explored exhaustive combinations between these parameters.

We have performed experiments with patches of different species in the same environment (see response to comment 2.3.a below). We have also clarified our definition of dimensionality reduction (see response to comment 2.3.b below).

2.1.b. Since the authors derive the rule of thumb by measuring end-point patch occupancy exclusively, it is then important to substantiate the claim that “patch occupancy to be roughly constant at the end of the (2 hour) experiment” (lines 68-70). No data is shown to support this important statement regarding the steady-state nature of patch occupancy in the authors’ experimental design.

We agree that this claim should be better substantiated. It was based on preliminary experiments where we recorded videos and studied the dynamics of patch occupancy. We have now added a supplementary figure showing results from these videos (Figure S), which is cited in line 87. We reproduce here Figure S1:

Figure S1. Dynamics of *C. elegans* foraging experiments. (A) Average number of worms in each patch (or outside the patches) as a function of time from two experiments with 4 food patches. Averaged over 58 videos, with 299 worms in total. Light patches show the 95% confidence intervals, computed via bootstrapping. Food patches were composed of *E. coli* OP50 with densities 0.2, 0.1, 0.05, and 0.025 (measured as Optical Density). Food patches were located at 1.2 cm of the center of the plate, forming a square. Worms started at the center of the square. All other experimental details were identical as those in the main text, except that the plates did not contain novobiocin (these experiments were performed with *E. coli* OP50 only, and this species' growth is effectively arrested by chloramphenicol alone). **(B)** Same as A, but for patch densities 0.2, 0.1, 0.025 and 0.0125 (measured as Optical Density), and averaged over 56 videos, with 266 worms in total.

2.1.c. There are several general statements that should be better qualified with “under our tested conditions”, example: “*C. elegans* response to all bacterial strains follows a universal one-dimensional trend” (lines 60-61), and “*C. elegans* response to food across bacterial species is driven by a single variable” (line 297-298). Also, the point about bacteria not growing and therefore not producing metabolites/toxins that may affect the natural foraging context (lines 320-321, 329-331) should be made more explicitly earlier, to qualify the statement regarding the lack of difference between bacterial strains in generating differential foraging response, especially if the authors are to make arguments regarding “ecological relevance” (line 332).

We have rewritten as follows:

Old lines 60-61 (line 75 in the new ms) now reads “Despite this high degree of complexity, *C. elegans*' response to all our bacterial strains followed a universal one-dimensional trend.”

Old line 297 (line 446 in the new ms) now reads “In our experiments, *C. elegans*' response to food across bacterial species was driven by a single variable”.

We have clarified earlier that bacteria could not grow on the plates. Lines 81-84 at the beginning of results now read “The food patches were placed on the day before the experiment, and we ensured that bacterial density remained unchanged by preparing the agar plates without nutrients and with a low dose of bacteriostatic antibiotics.”

2.1.d. The authors suggest that their high throughput assay allows them to obtain higher accuracy in behavioural measurements that may otherwise be hidden by behavioural variability (lines 41-42). The authors should then demonstrate such increase in accuracy, especially since variability is an inherent design principle for behaviour, so it is important to acknowledge the inherent variability and distinguish it from the noise that comes from the low throughput nature of most behavioural experiments.

By “high accuracy” we meant that having a high number of replicates allows to average out the variability, obtaining a highly reproducible average behavior. This high accuracy is demonstrated by our small errorbars and smooth experimental plots, and we have clarified what we mean by accuracy by rewriting the old lines 41-42, which now read as follows (lines 58-60 in the new ms): “Reaching at the same time a high number of combinations and a sufficient number of replicates to obtain highly accurate average behavior is beyond the experimental throughput in most behavioral experiments.”

2.2. Data reporting and availability

2.2.a. The figure legends should include explicit n numbers for each condition. The patch foraging experiments only report aggregate replicate numbers in the methods section: up to 32 replicates per condition for patch foraging (lines 445-452). The fitness experiments don't appear to report n numbers in the figure legend nor in the methods section.

We have added Table S1 with the sample sizes of all experiments, and we now cite this table from all figure captions.

2.2.b. The analysis code should be made publicly available. Given the small size of the dataset (~2000 still images), perhaps the full dataset could also be shared.

We have added the full dataset and the Matlab code that generates all the figures in the paper as Supplementary Data 1. This code generates the figures from almost raw data (number of worms in each patch in each plate).

2.3. Conceptual

2.3.a. The authors show mixed food patch experiments (page 10) using pre-mixed food combination at different densities to demonstrate that their rule of thumb holds. A different experiment that would conceptually strengthen their argument further is to use different bacteria type for the patches, at different densities, in a single foraging experiment. This would really demonstrate the dimensionality reduction by showing that the food type does not matter, since as mentioned before (see 1a above), true bacteria type x density combinations in distinct patches have not yet been explored in this work despite the authors' claim of exhaustive mapping of these in their abstract.

Thank you for this suggestion. We have performed this experiment, and it matched with our predictions. This experiment is shown in a new Figure 4, new Figure S6, and lines 363-373, which we reproduce here:

“All previous results correspond to experiments in which worms were exposed to a single bacterial strain at a time. We next tested if our one-dimensional rule could predict patch occupancy in environments containing food patches of several different species. To test this, we used *Escherichia coli* OP50 plus two other randomly chosen strains, which we chose randomly just making sure that the three strains covered a wide range of D_{attract} (DA1880 and CR266) . We then performed experiments with five patches of different species and at different densities (**Figure 4a**), recorded the number of worms at each food patch after 2 hours, and compared these results with the results predicted by our one-dimensional model (using the same parameters as in **Figure 1f**). We performed this experiment for many different combinations of densities of the three bacterial strains (**Figure S6**), obtaining a good overall agreement between predictions and experimental results (**Figure 4b,c**).”

Figure 4: Results in environments with food patches of different bacterial species. **a.** Schematic of the experimental set-up: We placed 5 food patches of different densities and different bacterial species, and counted the number of worms per patch after 2 hours. **b.** Relative number of worms found at each food patch, as a function of effective density ($D/D_{attract}$). Black line: Sigmoid with $H = 146$, $k = 1.4$. **c.** Measured proportion of worms in each food patch, versus proportion predicted by the sigmoid using effective density and the same parameters as in box b. See **Figure S6** for results separated by condition, **Table S1** for sample sizes and **Supplementary Data 1** for the data and computer code that generate this figure. All errorbars are 95% confidence intervals, computed via bootstrapping.

2.3.b. The authors should unpack a bit more the concept of “information bottleneck” (lines 105-107), how this necessarily relates to a small nervous system, what this means in ecological and evolutionary terms, since these concepts are central to their proposed theory for why a rule of thumb may exist and what we may learn from it.

We have expanded this concept. Lines 128-164 now read as follows:

“We now focus on the dimensionality of the rule we found. We define dimensionality as the number of variables needed to describe how behavior changes in response to environmental change. Complex environmental changes involve many variables. For example, two food patches may differ in bacterial density, shape, size and hardness of the bacteria, viscosity, presence of biofilms, concentration of a myriad of metabolites secreted by the bacteria, composition of the bacteria themselves, etc. We hypothesize that there is a bottleneck in the animal’s information processing system, so that the many parameters that may affect the animal’s sensory inputs are transformed in a few parameters that determine the behavior. A model that exploits this bottleneck will have few parameters that depend on the environment, so we define the dimensionality as the number of parameters that need to be re-fit when the environment changes. This definition is not the same as the total number of parameters of the model, because hard-wired parameters don’t change when the environment changes, and therefore don’t count towards the dimensionality of the model. While our method of analysis is purely behavioral and does not intend to describe *C. elegans*’ neural computations, the discovery of a low-dimensional rule that describes a wide range of experimental conditions would suggest an information bottleneck in *C. elegans*’ nervous system, and this result would be in line with previous works that found a low-dimensional manifold in *C. elegans*’ brain dynamics⁴⁰.”

2.3.c. How do we reconcile between the small nervous system and the complex foraging behaviour (lines 51-53) that co-exist in *C. elegans*, in the light of the authors’ findings? How do the authors’ finding fit in with the other thoroughly developed existing foraging theory? (line 48) These points should further discussed.

Most of the complexity in foraging behavior described in previous literature pertains the mechanisms that underlie our results. While we find a simple pattern in patch occupancy, we have not studied how that patch occupancy is reached (e.g. the role of exploration, chemotaxis, etc.). We have added a paragraph discussing this in lines 471-485, which reads as follows:

It remains an open question what mechanisms are responsible for *C. elegans*’ response to bacterial cell number. A possible hypothesis would be that bacterial number is the easiest way for *C. elegans* to estimate bacterial density. *C. elegans*’ seems to estimate food availability from the amount of ingested through its grinder per unit time,^{34,45} and this measurement might be driven by the number of discrete bodies swallowed. However, this hypothesis would not explain why *C. elegans* lays more eggs when feeding on high cell density food patches, regardless of their biomass density. Two alternative hypotheses would be consistent with this result: First, it might be that a key nutrient is produced by the bacteria on a cell-by-cell basis, and that both *C. elegans* attraction to food and its egg-laying capability correlate with the concentration of this nutrient. Second, it might be that high cell number makes it

easier for *C. elegans* to find and consume bacteria. This might be especially important at low bacterial densities: At a density of, for example, 10^4 cells/mm², only about 1% of the surface of a food patch is actually covered by bacteria. Therefore, at this density worms must find individual cells or small clusters of cells, which are sparsely distributed. The success in this search might be the key factor driving *C. elegans*' feeding rate, and it would depend on cell number rather than on total biomass density. We currently have insufficient evidence to decide between these two hypotheses.

2.3.d. This is probably beyond the scope of this paper, but I wonder what happens when 20, instead of 10 worms are used for the patch foraging experiments? Does the rule of thumb still apply, or does the effect of social foraging come into play at some critical threshold?

We believe that the exact number of worms does not matter, as long as it's below the threshold where strong collective effects arise. We have discussed this question further in lines 87-90, which we reproduce here:

“We placed approximately 10 worms per plate, and discarded any plates with more than 20 worms. We kept this low number of individuals per plate to prevent food depletion, and also to prevent collective effects such formation of clusters^{37,38}, or networks³⁹, which require much higher worm densities (hundreds to millions of worms per plate).”

2.4. Technical

2.4.a. Brood size is typically used as a fitness measurement and the full brood period is over 3-4 days for N2 *C. elegans* worms. The authors measured the number of eggs laid over five hours in Day 2 adults at the height of their typical brood period, and admit that this is a proxy and not a full measurement of fitness. There could be other explanations for why eggs laid during this five hours may not necessarily correlate with the overall brood size/fitness. For example it is conceivable that certain food conditions may make the worms lay more eggs, or lay eggs faster, on Day 2 but have overall fewer eggs over the full brood period. The authors should discuss such possibility, and perhaps support their methodology by providing an intuitive explanation for why the five-hour egg laying activity is still a useful read-out for fitness: perhaps it is a good strategy to lay more eggs on better food because the eggs produced on these food may be better quality, etc.

We agree with this point. We were not explicit enough with the fact that we are not actually interested in overall fitness or brood size, since this is a long-term metric. Instead, we are interested on the impact on fitness of a relatively short period of exploitation of a given food patch, because this is what should drive the stay/leave decision. This impact on fitness must necessarily be estimated with a short-term proxy.

We have been more careful to use “impact on fitness”, or “egg-laying” rather than just “fitness” or other expressions throughout the text (especially in lines 239-270). Also, we have added a paragraph in the discussion to explain this. This paragraph reads as follows (lines 527-543):

“A third limitation of our study is the use of egg-laying as a proxy to fitness. Fitness is an elusive magnitude, and measuring it directly is hard, because it requires long-term measurements over

several generations^{49,50}. Different proxies of fitness can be used, and previous studies in *C. elegans* used offspring number^{51,52}, development rate^{27,51}, number of offspring that reach adulthood^{51,53} longevity⁵¹, and others⁵⁴. While longer-term measurements are more reflective of the actual evolutionary course of the population, they only make sense when the environment remains relevant and stable over a sufficiently long period of time. This was not our case, since our fitness experiments were performed with a single food patch, rather than in a more naturalistic environment. These conditions forced us to use egg-laying as a short-term proxy of fitness, with the additional limitation that we could not wait for eggs to hatch, which might further reduce the accuracy of our measurements (see Methods). However, it is worth noting that the factors affecting long-term fitness in a naturalistic environment (such as future availability of food, future temperature, predation, etc.) are uncertain while an individual is exploiting a food patch. Therefore, foraging decisions are probably driven by the instantaneous benefit of the food on the worm's ability to produce offspring, which is well represented by the number of eggs measured in our experiment."

Also, we have added a paragraph in Methods explaining why we counted unhatched eggs rather than hatched eggs. This paragraph reads as follows (lines 769-774):

"We counted unhatched eggs (rather than waiting for them to hatch, which would also take into account viability) because of experimental limitations. Our experiment required more than 1500 plates, so it would have been impractical to remove manually every adult after 5 hours to allow for more time for the eggs to hatch. Also, the copper ring was not 100% effective at preventing worms from escaping. While it was easy to discard all cases in which the adult had escaped, missing larvae would have decreased the accuracy of our measurements."

2.4.b. Food preference in a foraging context typically refers to preferential choice of one type of food over another, i.e. different bacteria strains. Such food choice experiments were not the scope of this paper and thus I am not sure it is appropriate to compare food preference between different strains by looking at Dattract values that were obtained for single bacteria strain at one time, at different density conditions. Perhaps the authors could better define or clarify their use of terminology "preference" (line 165) and "food choice" (lines 210-211) in their context for disambiguation. How is exactly "preference" (line 182) read from the plot in Figure 2d? "Neglect" (line 184) and "probably neglecting" (line 189) also reads too strong for alternative explanations.

We have performed experiments with several species in the same plate (see our response to point 2.3.a above).

Also, we have rewritten the text to be more careful with this:

The old line 165 now reads "Attraction to a food patch correlates with its impact on fitness" (line 239 in the new ms).

We have kept "food choice" unchanged at the old lines 210-211 (line 317 in the new ms). This sentence is a general statement about what properties of food determine *C. elegans*' response, and we didn't find a better term than "food choice".

We have clarified the old line 182 (line 261 in the new ms), which now reads “This previous experience might explain the deviation, because *C. elegans* can learn odors and tastes associated with beneficial food,^{21,23} and this learning might increase the attractiveness of OP50 with respect to unfamiliar strains.”

We have reworded the old line 184 (line 263 in the new ms), which now reads “The other two outliers (*Bacillus safensis* CR164 and *Pseudomonas viridiflava* CR90) cannot be explained in this way, and are probably due to factors that impact fitness but have weaker influence on the behavior of the worms.”

For the sake of concision, we have kept “probably neglecting” in the old line 189 (line 270 in the new ms).

2.4.c. Under this set of experimental design, cell density presented as cells/mm² and cells/unit volume are scalable with each other since the surface area of the patch is roughly the same for their experiments. However, one can imagine a situation where the same volume leads to a different surface area, for example because the surface tension of the hydrogel changes and the spotted liquid food spreads to a bigger patch with a larger surface area. Then would the key variable that worms respond to be cell density as measured by unit surface or by unit volume?

We believe that surface density is the relevant variable, since the worms have no access to the volume density before spotting the drop, but we don't have data to support this belief. We did not find any significant difference in the surface of drops for different bacterial strains, and we don't have any data where we manipulate surface tension. Given that we can only speculate about this question, we preferred to not discuss it explicitly in the text, but our new paragraph in lines 471-485 discussing potential mechanisms may give some insight.

2.4.d. Lines 86-87: what does a “control patch without bacteria” look like and how was this control performed?

In some of our experimental conditions, we left one of the 5 positions in the pentagon without any drop, but we then counted the number of worms in this area as if there was a food patch. These are what we called “control patch without bacteria”. We realize that the term “control patch” makes it seem as if it's a separate experiment, so we have rewritten this sentence to make it clearer (line 114), and now it reads: “...the number of worms found at patches with zero density”

We have also clarified this in the Methods, where we have added the following (lines 636-639): “Some of our experimental conditions contained a food patch with zero density. In these cases, we simply left one of the drop positions empty (we did not pipet anything on that position), but then counted the number of worms in that area as if there was a food patch.”

2.5. Minor comments:

2.5.1. The authors mentioned some environmental variables such as spatial arrangement (line 35) that were not tested, presumably since the patches are randomly positioned on the vertices of the pentagon. Given the high n number for the patch foraging experiments (up to 32

replicates per food type), perhaps the effect of spatial arrangement is also possible to explore with the existing dataset?

Thank you for this suggestion. Unfortunately, our 32 replicates don't seem sufficient to give a good answer to this question. We typically had 6 randomizations of the relative positions of the food patches, so for each position we only had 5-6 replicates. Also, given that we randomized the relative positions without following any logic, we don't have a systematic set of relative position changes. For these reasons, when we looked at differences driven by changes in the relative positions, we did not find any clear result. In any case, we found at most weak effects from changing the relative positions, which reassured us that our qualitative conclusions would not be affected by any such effects. And while we cannot rule out that some effect exists, it's too weak to reach useful conclusions with our available data.

2.5.2. The point of determining whether biomass or cell density is the key factor for determining effective density can be better introduced (line 191). Right now these two possibilities read as if they are additional environmental variables to, rather than alternative explanations for, effective density. Also, these are properties of the food and I would suggest rewording them as such, rather than "sensory cues" (line 191) since the authors do not directly explore sensory input onto the nervous system per se. "Amount of food eaten should be the main driver of foraging behavior" (line 191-192) requires a citation and seems a bit out of place here to be stated a priori.

We have reworded the introduction to this section, which now reads as follows (lines 272-275):

"We next asked what properties of the food determine the observed rule of thumb. We first hypothesized that worms would choose the food patches with the highest biomass density, since biomass density determines the actual amount of food available. So far we have reported bacterial density using optical density (OD), which measures the amount of light absorbed by a bacterial culture."

2.5.3. Typos

- a. Line 49: "high offspring" ◇ high offspring number
- b. Line 65: "bacteriophage" ◇ bacterivore
- c. Line 223: "collapse"?
- d. Line 275: "run"◇ ran
- e. Line 346: "48-old" ◇ "48-hour old"

Corrected, thank you very much!

Reviewers' comments:

Reviewer #1 (Remarks to the Author):

The authors have addressed my concerns in my previous comments. However, after reading the manuscript again, I noticed a few important points that I had overlooked, which could drastically affect the conclusion of the work.

Major drawbacks:

1. In Figures 1d and 1f, the text above reads "Rule with 4 dimensions" and "Rule with 1 dimension", as if the plots in Figure 1f only have 1 free parameter to fit. However, in reality, the parameters H and k are also free and need to be determined to find the fitted black line in Figure 1f. Moreover, the parameters D and D_{attract} were lumped together to form a new parameter D/D_{attract} , as if it were a single dimension that *C. elegans* explores. However, it was more of a normalization than a reduction of the dimensions, since both D (food abundance) and D_{attract} (the "quality" of the food source) are independently important to determine the preferences of *C. elegans*. The fact that hundred-fold differences in D are now removed by dividing by D_{attract} implies that D_{attract} is indeed an important parameter to determine. Hence, none of the dimensions presented in Figure 1d have been reduced in Figure 1f.

2. In Figure 3a, the authors showed that the correlation between biomass density at OD = 1 and attraction density was poor. They further argued that the differences in biomass among bacteria are a poor determinant of food preference since the relation between biomass at OD=1 and the attraction density plot does not have a slope of 1. However, this is a missing link between the reasoning and the conclusion. To rebut (or establish) that biomass density at OD = 1 is an important parameter for the *C. elegans*, the authors should plot the relative number of worms vs. D, with or without normalization by the biomass density at OD = 1.

Minor point regarding presentation:

1. As the four parameters are key to the understanding of Figures 1d and 1f, the authors should present their mathematical model in the main text. Although not essential, there are more elegant forms of sigmoid relations available. For instance, a baseline adjustable form of the Hill equation. The authors should take advantage of such mathematical formulations, as they are better studied and the parameters in such models may be more easily understood by the readers.

2. There are a few word redundancies. The authors should check their text more carefully.

Reviewer #2 (Remarks to the Author):

Following the authors' revisions I feel that the manuscript has much improved. The additional experiments and supplementary information have helped to convince me of their major claims pertaining to the behavioural rule of thumb, the premise and the result of which I find exciting.

I still have some reservation about the fitness measurements. The authors have added text explaining both the conceptual and the technical reasons behind using an egg count from the first five hours of the experiment so I am now more convinced, but I believe that the age of the worm is very critical here for this fitness measurement assay and perhaps the authors can edit/expand their text to elaborate on this more. I had initially misunderstood the age of the worm for which the assay was carried out—when the authors reported "48-hour old" young adult worm they meant 48 hours post-L1 refeeding at 22 degrees, not 48 hours into their adulthood. Presumably at this age the worms are just at the very beginning of their brood period instead of mid-way through it. This makes a big difference when assessing partial brood over a chosen window, so the main text about the age of the worm

should more explicitly explain this without needing to go into supplementary information. Moreover, in my experience when bleach synchronising worms not all worms exit L1 diapause simultaneously to continue development, so there are some variations in the age of the worm up to a few hours. For a five-hour egg laying window at the beginning of the brood period, this source of variation would seem very critical for the collection and interpretation of the results, and I would appreciate if the authors could account for this in their manuscript.

I also have a comment on the conceptual link between information bottleneck in a small nervous system and the low dimensional manifold of whole brain dynamics that the authors have now added (lines 163-164). I had understood the latter to be related to global brain states and locomotion modes; the link to neural processing and bottleneck is less obvious to me so I would appreciate some further unpacking and elaboration if the authors were to make this connection.

Finally, some further minor editing is still needed for the manuscript. Here's a non-exhaustive list:

- lines 28-32: the reference to existing results but using the language of "seems to", "might drive", "could rely", "would yield" reads a bit strange to me
- lines 32-35 are repeated with different citations
- line 69 do the authors mean to refer to the adaptive value/fitness consequence by describing "well adapted" behaviors?
- line 115 and 637: zero [bacterial] density to distinguish from worm density
- lines 133-174: this chunk of text could be further edited for clarity regarding fixed vs. free parameters
- Fig 2b: the legend refers to "solid horizontal line" which for me first guides the eye to the error bars surrounding the solid black dot. Perhaps using a different colour for the "no food" horizontal line will help point to it more easily?
- Line 265 referring to Fig 2d: how come "probably" this explanation instead of the other way around, especially when the two outliers occupy opposite sides of the plot?
- Lines 367-368: "randomly chosen" is repeated, but also not random – this sentence needs editing and perhaps also refer to the supplementary figure where the D attract values are
- Line 473 and 493: apostrophe not needed here and in a few other places
- Line 477: "nutrient" or molecule?
- Line 594: grow grown
- Fig S3: there are two 1-patch experiments but a total of four blue box plots in two shades of blue. It took me a good while before I understood for 1-patch experiments there are two data points per experiment, for 0 and 1 ratio. The legend could be better written for clarity regarding this.

NOTE: All line numbers refer to the PDF version with tracked changes

Reviewers' comments:

Reviewer #1 (Remarks to the Author):

The authors have addressed my concerns in my previous comments. However, after reading the manuscript again, I noticed a few important points that I had overlooked, which could drastically affect the conclusion of the work.

Major drawbacks:

1. In Figures 1d and 1f, the text above reads "Rule with 4 dimensions" and "Rule with 1 dimension", as if the plots in Figure 1f only have 1 free parameter to fit. However, in reality, the parameters H and k are also free and need to be determined to find the fitted black line in Figure 1f. Moreover, the parameters D and D_{attract} were lumped together to form a new parameter D/D_{attract} , as if it were a single dimension that *C. elegans* explores. However, it was more of a normalization than a reduction of the dimensions, since both D (food abundance) and D_{attract} (the "quality" of the food source) are independently important to determine the preferences of *C. elegans*. The fact that hundred-fold differences in D are now removed by dividing by D_{attract} implies that D_{attract} is indeed an important parameter to determine. Hence, none of the dimensions presented in Figure 1d have been reduced in Figure 1f.

After considering this issue, we realize that our definition of dimensionality is non-standard and may be confusing, and that our main conclusions can be presented without resorting to it. For this reason, we have decided to present our results without using the concept of dimensionality. The results and their interpretation remain unchanged, but we have simplified the language in the following ways:

We have updated the title, which now reads "A taste for numbers: *Caenorhabditis elegans* foraging follows a simple rule of thumb" (line 1). We have updated the abstract (lines 14-25) and one paragraph in the introduction (lines 34-82), and we have removed the discussion about dimensionality from results (paragraphs removed around lines 148 and 159, plus short references about dimensionality removed throughout the ms). Finally, we have removed the titles about dimensionality from Figures 1 and 3. Therefore, our revised manuscript does not make any claims about a specific number of dimensions of the model, but our conclusions remain the same: All experimental data collapse in a single sigmoid when we define an effective density (D/D_{attract}), and the strongest determinant of this effective density seems to be cell density.

2. In Figure 3a, the authors showed that the correlation between biomass density at OD = 1 and attraction density was poor. They further argued that the differences in biomass among bacteria are a poor determinant of food preference since the relation between biomass at OD=1 and the attraction density plot does not have a slope of 1. However, this is a missing link between the reasoning and the conclusion. To rebut (or establish) that biomass density at OD = 1 is an important parameter for the *C. elegans*, the authors should plot the relative number of worms vs. D, with or without normalization by the biomass density at OD = 1.

We have added Supplementary Figure S6, which includes the required plot and compares directly the results when using Optical Density and Biomass Density to predict patch occupancy using a single sigmoid. We basically see no difference between Optical Density and Biomass Density (R^2 goes from 0.76 to 0.77), confirming our conclusion. We reproduce here Figure S6:

Figure S6. Comparison of results when using different measurements of density as the single environmental variable. Color and shape of markers identify bacterial strains (see legend in **Figure 1**). **a.** Relative number of worms found at each food patch, as a function of bacterial density (measured in OD) in the food patch. Points coming from the same experimental condition are linked by lines. Black line: Sigmoid, fitted to all strains. **b.** Measured proportion of worms in each food patch, versus proportion predicted by the sigmoid in (a). All errorbars show the 95% confidence interval, computed via bootstrapping. **c.** Same as (a), but with bacterial density measured as biomass density. **d.** Same as (b), but with using biomass density to perform the prediction.

Minor point regarding presentation:

1. As the four parameters are key to the understanding of Figures 1d and 1f, the authors should present

their mathematical model in the main text. Although not essential, there are more elegant forms of sigmoid relations available. For instance, a baseline adjustable form of the Hill equation. The authors should take advantage of such mathematical formulations, as they are better studied and the parameters in such models may be more easily understood by the readers.

Thank you for the suggestion. In this case we would prefer to keep the original version of the manuscript, for the benefit of non-technical readers. Our paper is intended to a wide audience, including experimentalists who may not be interested in the technical details of the fit. We believe that the visual description of the parameters in Figure 1c gives enough information for most readers, and having the sigmoid as the first equation in Methods makes it accessible enough for more technical readers, while keeping the main text clear and focused on our main conclusions.

About the choice of the sigmoid, during the preparation of the manuscript we tested different sigmoidal functions, and we found this one to be optimal in terms of goodness of fit, robustness and simplicity. For example, some sigmoidal functions gave an equally good fit, but their parameters were not so easily interpretable or were too sensitive to the type of experimental noise that we have in our data.

2. There are a few word redundancies. The authors should check their text more carefully.

Thank you for pointing this out, and sorry for the mistakes. We have revised the whole manuscript and removed all redundancies.

Reviewer #2 (Remarks to the Author):

Following the authors' revisions I feel that the manuscript has much improved. The additional experiments and supplementary information have helped to convince me of their major claims pertaining to the behavioural rule of thumb, the premise and the result of which I find exciting.

I still have some reservation about the fitness measurements. The authors have added text explaining both the conceptual and the technical reasons behind using an egg count from the first five hours of the experiment so I am now more convinced, but I believe that the age of the worm is very critical here for this fitness measurement assay and perhaps the authors can edit/expand their text to elaborate on this more. I had initially misunderstood the age of the worm for which the assay was carried out—when the authors reported “48-hour old” young adult worm they meant 48 hours post-L1 refeeding at 22 degrees, not 48 hours into their adulthood. Presumably at this age the worms are just at the very beginning of their brood period instead of mid-way through it. This makes a big difference when assessing partial brood over a chosen window, so the main text about the age of the worm should more explicitly explain this without needing to go into supplementary information. Moreover, in my experience when bleach synchronising worms not all worms exit L1 diapause simultaneously to continue development, so there are some variations in the age of the worm up to a few hours. For a five-hour egg laying window at the beginning of the brood period, this source of variation would seem very critical for the collection and interpretation of the results, and I would appreciate if the authors could account for this in their manuscript.

This is an excellent point, thank you. We agree that the age of our worms is not the ideal one for egg-laying quantification, and that our results may be noisier as a result. Our experimental protocol

requires young worms, because it would be hard to remove the offspring of older worms in our behavioral experiments.

Fortunately, this limitation does not change our conclusions. The age of the worms may increase the noise in our measurements, but it does not introduce any systematic bias thanks to a thorough randomization of our experimental conditions. The increased noise did not prevent us from reaching our conclusions, which would probably be stronger with cleaner measurements. We believe that the 95% confidence intervals that we computed for every egg-laying measurement and for the density threshold D_{fitness} (shown in Figure 2) provide a fair representation of the experimental variability.

We have clarified the age of the worms in the results section (line 106) and we have clarified our age criterion in Methods (line 565). We have also added a discussion of this experimental limitation (lines 767-774).

I also have a comment on the conceptual link between information bottleneck in a small nervous system and the low dimensional manifold of whole brain dynamics that the authors have now added (lines 163-164). I had understood the latter to be related to global brain states and locomotion modes; the link to neural processing and bottleneck is less obvious to me so I would appreciate some further unpacking and elaboration if the authors were to make this connection.

Given that reviewer 1 expressed doubts about our definition of dimensionality, and after consulting with other colleagues, we have decided to remove the concept from the paper. All our results and conclusions remain unchanged, but we have simplified the language, and this comment about the information bottleneck has been lost in the process (see our response to the first comment of reviewer 1).

Finally, some further minor editing is still needed for the manuscript. Here's a non-exhaustive list:

- lines 28-32: the reference to existing results but using the language of "seems to", "might drive", "could rely", "would yield" reads a bit strange to me

We have edited this.

- lines 32-35 are repeated with different citations

Thank you, we have removed the duplication.

- line 69 do the authors mean to refer to the adaptive value/fitness consequence by describing "well adapted" behaviors?

Yes, we did. This sentence meant to say that optimal foraging manifests across several different basic behaviors, including exploration, learning and feeding. We have rewritten the sentence to improve clarity (line 91).

- line 115 and 637: zero [bacterial] density to distinguish from worm density
Corrected.

- lines 133-174: this chunk of text could be further edited for clarity regarding fixed vs. free parameters

As discussed above, this text has now been removed.

- Fig 2b: the legend refers to “solid horizontal line” which for me first guides the eye to the error bars surrounding the solid black dot. Perhaps using a different colour for the “no food” horizontal line will help point to it more easily?

We have replaced it by a dotted line to prevent this confusion.

- Line 265 referring to Fig 2d: how come “probably” this explanation instead of the other way around, especially when the two outliers occupy opposite sides of the plot?

We have removed this phrase. We have realized that it was confusing, and its message was basically a tautology. The sentence now simply reads “The other two outliers (*Bacillus safensis* CR164 and *Pseudomonas viridiflava* CR90) cannot be explained in this way.” (line 276).

- Lines 367-368: “randomly chosen” is repeated, but also not random – this sentence needs editing and perhaps also refer to the supplementary figure where the D attract values are

We have corrected this sentence (line 357).

- Line 473 and 493: apostrophe not needed here and in a few other places

Thank you, we have corrected them.

- Line 477: “nutrient” or molecule?

In this case, we prefer the word “nutrient”.

- Line 594: grow grown

Corrected, thank you!

- Fig S3: there are two 1-patch experiments but a total of four blue box plots in two shades of blue. It took me a good while before I understood for 1-patch experiments there are two data points per experiment, for 0 and 1 ratio. The legend could be better written for clarity regarding this.

Thank you for pointing it out. We have explained this in the figure caption (lines 1053-1058):

REVIEWERS' COMMENTS:

Reviewer #1 (Remarks to the Author):

The authors have addressed all my concerns. I found the manuscript to be interesting and well-supported with experimental evidence. Thank you.

Reviewer #2 (Remarks to the Author):

The revised manuscript addressed my major concerns.

Minor suggestion:

Line 106: (48-hour old)  (48-hour post-L1) to indicate the age of the worm.

Line 768: "48-old" copy editing error.

Line 112: Although references 36+37 that the authors cited do involve very high density worms, clusters and collective effects can be seen in population sizes well below hundreds so the sentence reads misleading and should be edited.

REVIEWERS' COMMENTS:

Reviewer #1 (Remarks to the Author):

The authors have addressed all my concerns. I found the manuscript to be interesting and well-supported with experimental evidence. Thank you.

Thank you!

Reviewer #2 (Remarks to the Author):

The revised manuscript addressed my major concerns.

Minor suggestion:

Line 106: (48-hour old)  (48-hour post-L1) to indicate the age of the worm.

We would prefer to leave this unchanged. We fear that “48-hour post-L1” might be misleading, since it might be interpreted as 48 hours after the molt between L1 and L2. A possibility would be to write “48-hour after re-feeding at the L1 stage”, but we feel that this is too technical for this point in the main text. Our criterion to define age is now fully explicit in Methods, so we believe it’s best to leave a less technical comment in the Results section.

Line 768: "48-old" copy editing error.

We didn’t find any error here. Perhaps it was an issue with the pdf conversion?

Line 112: Although references 36+37 that the authors cited do involve very high density worms, clusters and collective effects can be seen in population sizes well below hundreds so the sentence reads misleading and should be edited.

This sentence now reads “...to prevent collective effects such formation of clusters^{36,37}, or networks³⁸, which require higher worm densities”